# BINARY-FEEDBACK ACTIVE TEST-TIME ADAPTATION

## ABSTRACT

Deep learning models perform poorly when domain shifts exist between training and test data. Test-time adaptation (TTA) is a paradigm to mitigate this issue by adapting pre-trained models using only unlabeled test samples. However, existing TTA methods can fail under severe domain shifts, while recent active TTA approaches requiring full-class labels are impractical due to high labeling costs. To address this issue, we introduce a Binary-feedback Active Test-Time Adaptation (BATTA) setting, which uses a few binary feedbacks from annotators to indicate whether model predictions are correct, thereby significantly reducing the labeling burden of annotators. Under the setting, we propose BATTA-RL, a novel dual-path optimization framework that leverages reinforcement learning to balance binary feedback-guided adaptation on uncertain samples with agreement-based self-adaptation on confident predictions. Experiments show BATTA-RL achieves substantial accuracy improvements over state-of-the-art baselines, demonstrating its effectiveness in handling severe distribution shifts with minimal labeling effort.

## 1 INTRODUCTION

Deep learning has revolutionized various fields, including computer vision (Deng et al., 2009), speech recognition (Gulati et al., 2020), and natural language processing (Brown et al., 2020). However, deep models often suffer from domain shifts, where discrepancies between training and test data distributions lead to significant performance degradation. For example, autonomous driving systems might struggle with new types of vehicles or unexpected weather conditions that differ from the training data (Sakaridis et al., 2018).

Test-time adaptation (TTA) (Wang et al., 2021) is a viable solution to domain shifts by dynamically adopting the pre-trained models in real-time using only unlabeled test samples. However, without ground-truth labels, most TTA methods are vulnerable to adaptation failures (Gong et al., 2022; Niu et al., 2023; Gong et al., 2023b; Lee et al., 2024b). Recent studies showed that TTA failures are inevitable when there is significant divergence between test and training data (Press et al., 2024), especially in lifelong continual adaptation (Press et al., 2023).

To mitigate the issue, the paradigm of active TTA (Gui et al., 2024) was proposed, where an oracle (e.g., an annotator) provides ground-truth labels for a few selected samples during adaptation. However, obtaining such labels in real-world applications is often impractical due to its high cost and interaction bottlenecks, particularly when the number of classes is large. For example, full-class labeling by human annotators suffers a high labeling overhead (e.g., 11.7 sec per image) and a high labeling error rate (e.g., 12.7%) (Joshi et al., 2010). This necessitates a lightweight labeling approach to reduce the annotators' burden for TTA.

We introduce **Binary-feedback Active TTA (BATTA)** setting (Figure 1) where an annotator provides simple binary feedback on the model's predictions, indicating whether they are *correct* or *incorrect*. This approach only requires minimal label information, thereby significantly reducing labeling costs and mitigating interaction bottlenecks compared with full-label active TTA (Gui et al., 2024), making our framework more attractive for real-world TTA applications.

As a solution, we propose BATTA-RL, a dual-path optimization for BATTA that incorporates both binary feedback and unlabeled samples. Inspired by the recent reinforcement learning studies that show effectiveness in incorporating human feedback in the optimization process (e.g., RLHF, Ouyang et al. (2022)), BATTA-RL leverages reinforcement learning to effectively balance two complementary adaptation strategies: *Binary Feedback-guided Adaptation (BFA)* on uncertain samples and *Agreement-*

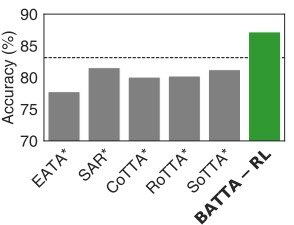

Figure 1: Overview of Binary-feedback Active TTA (BATTA) setting. Traditional TTA algorithms often fail under severe distribution shifts due to the fundamental risk of adapting to unlabeled test samples. Our proposed BATTA addresses this challenge by offering a few binary feedbacks (*correct* or *incorrect*) on selected model predictions. This approach significantly reduces labeling effort compared to full-class labeling while enabling robust adaptation.

Figure 2: Accuracy (%) of TTA baselines and BATTA-RL on CIFAR10-C. Notation * indicates a modified algorithm to utilize binary-feedback samples. The dotted line refers to full-labeled active TTA.

*Based self-Adaptation (ABA)* on confident samples (Figure 3). Using Monte Carlo dropout (Gal & Ghahramani, 2016) for policy estimation and uncertainty assessment, we select uncertain samples for binary feedback in BFA while leveraging samples with high prediction agreement in ABA. This dual approach enables BATTA-RL to adapt to new uncertain patterns (via BFA) while maintaining confidence in correct predictions (via ABA), therefore achieving robust performance improvements.

We evaluate BATTA-RL under BATTA setting with various test-time distribution shift scenarios, including three image corruption datasets (CIFAR10-C, CIFAR100-C, and Tiny-ImageNet-C) and two domain generalization scenarios (domain-wise and mixed data streams). Comparisons with TTA and active TTA methods demonstrate that BATTA-RL achieves an accuracy improvement of 11.9%p on average. Notably, BATTA-RL is the only method to outperform full-labeled active TTA under the BATTA setting (Figure 2). These results highlight the importance and effectiveness of BATTA-RL in addressing the BATTA problem, thereby enabling robust adaptation with minimal labeling effort.

## 2 BINARY-FEEDBACK ACTIVE TEST-TIME ADAPTATION

We propose Binary-feedback Active Test-Time Adaptation (BATTA), a test-time adaptation (TTA) setting where an oracle provides a few binary feedback (correct/incorrect) on the model prediction. BATTA addresses the critical challenge of adapting pre-trained models to domain shifts with minimal labeling effort. Unlike methods that require full-class labels, BATTA leverages simple binary feedback to guide the adaptation process. Specifically, full-class labeling is as expensive as $\log(\texttt{num\_class})$ times trial of binary-feedback labeling regarding the Shannon information gain (MacKay, 2003). Also, the human experiment of full-class labeling on 50-class showed 11.7 seconds of response time with a 12.7% error rate while comparing two images (analogous to our binary feedback approach) took only 1.6 seconds with 0.8% error rate (Joshi et al., 2010). These results demonstrate that object comparison (e.g., comparing an image with a model prediction in BATTA) requires a lower labeling overhead than full-class labeling, making BATTA more efficient and practical for real-world applications.

**Feedback mechanism.** In BATTA, an oracle provides a few binary feedbacks indicating whether the model's prediction is correct. This real-time feedback is integrated into the system, enabling continuous model adaptation. The binary feedback mechanism is illustrated in Figure 1, where the oracle evaluates the model's predictions, and the feedback memory is updated accordingly.

**Notation.** Let $x$ denote a test sample selected for active labeling at time $t$, and $y^* = \arg\max_y f_\theta(y|x)$ be the model's prediction with parameters $\theta$. The binary feedback $B(x, y)$ is given by:

$$B(x, y) = \begin{cases} 1 & \text{if } y \text{ is correct}, \\ -1 & \text{if } y \text{ is incorrect}. \end{cases}$$

As a result, binary-feedback active samples consist of $(x, y^*, B(x, y^*))$ with target instance $x$, model prediction label $y^*$, and binary feedback $B$.

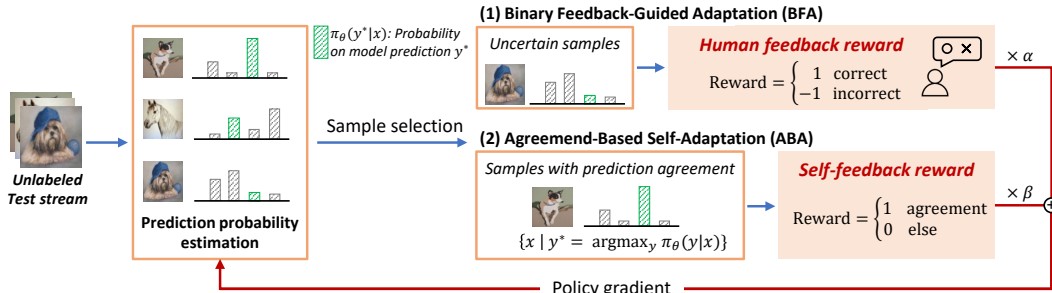

Figure 3: Overview of BATTA-RL algorithm. BATTA-RL formulates an RL-based dual-path approach for BATTA with prediction probability estimation via MC-dropout. We calculate the policy gradient with (1) binary feedback-guided adaptation on uncertain samples and (2) self-adaptation of unlabeled samples with prediction agreement.

## 3  BATTA-RL: DUAL-PATH OPTIMIZATION FRAMEWORK

**Motivation.**  Recent advancements in reinforcement learning with human feedback (RLHF, Ouyang et al. (2022)) have demonstrated the effectiveness of incorporating sparse feedback signals in large language model training. Inspired by this, we propose BATTA-RL, a reinforcement learning (RL) based approach for binary-feedback active test-time adaptation (BATTA) that effectively adapts to distribution shifts using minimal labeling effort. BATTA-RL leverages binary feedback as a reinforcement signal, offering several key advantages for test-time adaptation (TTA). (1) Binary feedback can be seamlessly incorporated as non-differentiable rewards in the RL framework, enabling the model to learn from minimal supervision (Zoph & Le, 2017; Yoon et al., 2020). (2) The RL framework allows for integrating binary feedback and unlabeled samples into a single objective function optimized through policy gradient methods. By combining sparse binary-feedback samples with unlabeled data, BATTA-RL provides a robust framework with minimal labeling effort, making TTA more feasible for real-world applications.

**Policy gradient modeling.**  Given a batch of test samples $\mathcal{B} = \{x_1, \ldots, x_n\}$, our goal is to adapt the model parameters $\theta$ to improve performance on the test distribution. We formulate the test-time adaptation process as an RL problem by assigning test-time input $x \sim \mathcal{B}$ as a state, the model prediction $y^* = f_\theta(x)$ as an action, and the corresponding prediction probability $\pi_\theta(y|x)$ as a policy, which objective is maximizing the expected reward, defined as:

$$J(\theta) = \mathbb{E}_{x \sim \mathcal{B}, y \sim \pi_\theta(y|x)}[R(x, y)], \tag{1}$$

where $R(x, y)$ represents the reward function defined later. This optimization is performed for each test batch, allowing continuous adaptation to the evolving test distribution.

As binary feedback is a non-differentiable function, we employ the REINFORCE algorithm (Williams, 1992), also known as the "log-derivative trick". This method allows us to estimate the gradient of the expected reward with respect to the model parameters:

$$\nabla_\theta J(\theta) = \mathbb{E}_{x \sim \mathcal{B}, y \sim \pi_\theta(y|x)}[R(x, y)\nabla_\theta \log \pi_\theta(y|x)]. \tag{2}$$

By using this gradient estimator, we can effectively optimize our model parameters using stochastic gradient ascent.

To estimate the policy $\pi_\theta$, we leverage Monte Carlo (MC) dropout (Gal & Ghahramani, 2016). MC-dropout approximates Bayesian inference by applying dropout at test time and performing multiple forward passes. This approach allows us to estimate the model's prediction probability without modifying the model architecture. Specifically, we approximate the policy $\pi_\theta(y|x)$ as:

$$\pi_\theta(y|x) = \frac{1}{N} \sum_{n=1}^{N} f_\theta^d(y|x), \tag{3}$$

where $f_\theta^d$ represents the model with dropout applied during inference, and $N$ is the number of forward passes.

With the proposed RL framework, BATTA-RL addresses the challenge of utilizing (1) *few samples with ground-truth binary feedback* and (2) *many unlabeled samples with potentially noisy predictions* through two complementary strategies:

1. Binary Feedback-guided Adaptation on uncertain samples (BFA, Section 3.1): This strategy focuses on enhancing the model's areas of uncertainty. By selecting samples where the model is least confident and obtaining binary feedback on these, BATTA-RL efficiently probes the boundaries of the model's current knowledge.

2. Agreement-Based self-Adaptation on confident samples (ABA, Section 3.2): To complement the guided adaptation strategy, BATTA-RL also leverages the model's existing knowledge through self-adaptation on confidently predicted samples. Without requiring additional feedback, ABA identifies confident samples by the agreement between the model's standard predictions and those obtained via MC-dropout.

The synergy between Binary Feedback-guided Adaptation (BFA) and Agreement-Based self-Adaptation (ABA) enables BATTA-RL to effectively utilize both labeled and unlabeled samples. BFA drives exploration and adaptation to new patterns in the test distribution through binary feedback on uncertain samples. Concurrently, ABA maintains and refines existing knowledge through self-supervised adaptation on confident predictions. This dual-path optimization allows BATTA-RL to adapt effectively across diverse challenging conditions.

### 3.1 BINARY FEEDBACK-GUIDED ADAPTATION ON UNCERTAIN SAMPLES

In BATTA settings where binary feedback is limited and costly, selecting samples for querying becomes crucial for effective model adaptation. To address this challenge, we propose Binary Feedback-guided Adaptation on uncertain samples (BFA). This approach refines the model's decision boundaries and improves its understanding of challenging data points through binary feedback guidance, enabling robust and efficient adaptation in test-time distribution shifts.

**Sample selection.** To guide the adaptation, we focus on the most uncertain samples, often the most informative for model improvement (Settles, 2009). We quantify the samplewise (un)certainty using MC-dropout, which we previously employed for policy estimation (Equation 3). MC-dropout offers a robust uncertainty estimate, while the standard softmax probabilities often exhibit overconfidence on out-of-distribution samples; a phenomenon observed in recent test-time adaptation studies (Gong et al., 2023b; Lee et al., 2024b). Therefore, we define the sample-wise certainty $C(x)$ as:

$$C(x) = \pi_\theta(y^*|x), \tag{4}$$

where $y^* = \arg\max_y f_\theta(y|x)$ is the current model prediction and $\pi_\theta(y|x)$ is MC-dropout softmax confidence.

To implement binary feedback-guided adaptation, we select the set of $k$ samples with the lowest certainty (i.e., highest uncertainty), noted as $\mathcal{S}_{\text{BFA}}$:

$$\mathcal{S}_{\text{BFA}} = \texttt{argsort}_x(C(x))[:k]. \tag{5}$$

**Reward function design.** For these selected samples, we query the binary feedback $B(x, y)$ (correct/incorrect) and define the reward function $R_{\text{BFA}}$ for active samples as:

$$R_{\text{BFA}}(x, y) = B(x, y) = \begin{cases} 1 & \text{if the prediction is correct,} \\ -1 & \text{if the prediction is incorrect.} \end{cases} \tag{6}$$

This binary-feedback reward scheme provides a clear signal for model adaptation, encouraging the prediction probability of correct predictions and penalizing incorrect ones. By selectively applying this reward function to the most uncertain samples, BFA efficiently utilizes the limited labeling budget, maximizing the contribution of each queried label.

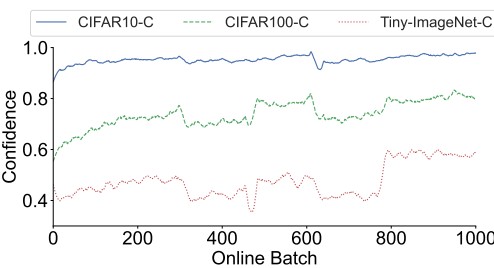 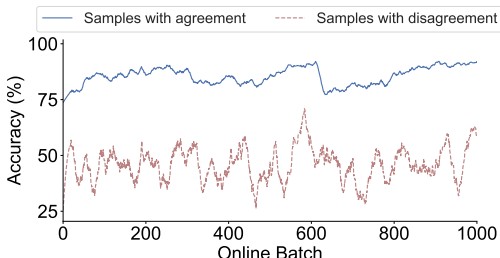

(a) Average samplewise confidence in online adaptation.

(b) Average samplewise accuracy of samples with agreement ($\mathcal{S}_{\text{ABA}}$) and disagreement ($\mathcal{B} \setminus \mathcal{S}_{\text{ABA}}$).

Figure 4: Analysis of confidence and accuracy during online adaptation. (a) Average sample-wise confidence over time and dataset, showing dynamic changes that challenge fixed thresholding methods. (b) Average sample-wise accuracy for samples with prediction agreement and disagreement on CIFAR10-C, demonstrating the effectiveness of agreement-based selection for confident samples.

### 3.2 AGREEMENT-BASED SELF-ADAPTATION ON CONFIDENT SAMPLES

To complement the binary feedback-guided adaptation on uncertain samples, we propose leveraging the model's confident predictions on the remaining many unlabeled samples. This approach, which we call Agreement-Based self-Adaptation (ABA), aims to reinforce the model's current knowledge without requiring additional oracle feedback.

**Sample selection.** The key idea behind ABA is to identify samples where the model's standard prediction agrees with its MC dropout prediction. We consider these samples "confident" and use them for self-adaptation. Formally, we define the set of confident samples $\mathcal{S}_{\text{ABA}}$ as:

$$\mathcal{S}_{\text{ABA}} = \{x \in \mathcal{B} \setminus \mathcal{S}_{\text{BFA}} \mid \arg\max_y f_\theta(y|x) = \arg\max_y \pi_\theta(y|x)\}, \tag{7}$$

where $\mathcal{B}$ is the entire batch of test samples, $\mathcal{S}_{\text{BFA}}$ is the set of samples selected for active feedback, $f_\theta(y|x)$ is the standard model prediction, and $\pi_\theta(y|x)$ is the MC-dropout prediction.

Unlike existing test-time adaptation (TTA) methods that rely on fixed confidence thresholds (Niu et al., 2022; 2023; Gong et al., 2023b), our approach can dynamically select confident samples based on the agreement between standard and MC-dropout predictions. Figure 4a illustrates the dynamic nature of prediction confidences during distribution shifts—necessitating the need for dynamic sample selection. To demonstrate the effectiveness of ABA further, we compare our agreement-based approach with various thresholding strategies in Figure 8 in Appendix B. The results support the superiority of our dynamic selection method over confidence thresholding.

Furthermore, our method effectively identifies confident samples for self-adaptation. Figure 4b demonstrates the stable accuracies in samples with agreement, while samples with disagreement show unstable and low accuracies. This originates from the prediction agreement of indicating robustness and reliability via the consistency in model outputs across different dropout masks. By leveraging this consistency, ABA can reliably select confident samples for effective self-adaptation.

**Reward function design.** We now incorporate these samples into our reinforcement learning framework. We introduce a self-feedback reward function $R_{\text{ABA}}$ for unlabeled samples. This reward encourages the model to maintain its predictions on confident samples while discarding the adaptation on unreliable ones. Formally, we define $R_{\text{ABA}}$ as:

$$R_{\text{ABA}}(x, y) = \begin{cases} 1 & \text{if } x \in \mathcal{S}_{\text{ABA}}, \\ 0 & \text{otherwise.} \end{cases} \tag{8}$$

By incorporating this adaptive prediction agreement strategy, ABA enhances the learning process by maintaining the knowledge of confident predictions. While prediction disagreement might suggest uncertainty, our analysis shows these samples exhibit mixed accuracy rather than consistent errors (Figure 4b). Therefore, ABA assigns zero rewards to disagreement cases rather than penalizing them (as in BFA), preventing potentially harmful adaptation from noisy signals. This conservative approach

is especially valuable in TTA scenarios where distribution shift may be partial or gradual, where most of the model's existing knowledge remains relevant.

### 3.3 BATTA-RL Algorithm

Our proposed BATTA-RL algorithm integrates binary feedback-guided adaptation (BFA, Section 3.1) and agreement-based self-adaptation (ABA, Section 3.2) into a unified dual-path optimization framework, enabling effective adaptation to distribution shifts while maintaining model stability.

To achieve this, we formulate a combined objective function that balances the rewards from both uncertain samples (guided by binary feedback) and confident samples (identified through prediction agreement). Formally, we define our total objective function $J_{\texttt{total}}()$ as:

$$J_{\texttt{total}}(\theta) = \alpha \mathbb{E}_{x \in \mathcal{S}_{\texttt{BFA}}}[R_{\texttt{BFA}}(x, y)] + \beta \mathbb{E}_{x \in \mathcal{B} \setminus \mathcal{S}_{\texttt{BFA}}}[R_{\texttt{ABA}}(x, y)], \tag{9}$$

where $\alpha, \beta$ are hyperparameters to control the relative contributions of BFA and ABA.

Following the REINFORCE algorithm, the gradient of our total objective is given by:

$$\nabla_\theta J_{\texttt{total}}(\theta) = \alpha \mathbb{E}_{x \in \mathcal{S}_{\texttt{BFA}}}[R_{\texttt{BFA}}(x, y)\nabla_\theta \log \pi_\theta(y|x)] + \beta \mathbb{E}_{x \in \mathcal{B} \setminus \mathcal{S}_{\texttt{BFA}}}[R_{\texttt{ABA}}(x, y)\nabla_\theta \log \pi_\theta(y|x)]. \tag{10}$$

This gradient estimation guides our parameter updates via stochastic gradient ascent, refining the model's performance on the evolving test distribution. We utilize a single value of $\alpha = 2$ and $\beta = 1$ for all experiments. Further details are in Appendix D.

## 4 Experiments

We present our experimental setup and the results across various scenarios in BATTA setting. We evaluate the performance of BATTA-RL against state-of-the-art baselines, ensuring fairness by providing an equal amount of binary-feedback active samples. Additional experiments, additional results, and experiment details are provided in Appendices B, C, and D.

**Baselines.** We evaluated BATTA-RL against a comprehensive set of baselines, including source validation (SrcValid) and seven state-of-the-art TTA methods: BN-Stats (Nado et al., 2020), TENT (Wang et al., 2021), EATA (Niu et al., 2022), SAR (Niu et al., 2023), CoTTA (Wang et al., 2022), RoTTA (Yuan et al., 2023), and SoTTA (Gong et al., 2023b). To ensure a fair comparison, we incorporate an equal number of random binary-feedback data into TTA baselines by adding correct-sample loss (cross-entropy) and incorrect-sample loss (complementary label loss from Kim et al. (2019)). Additionally, we included SimATTA (Gui et al., 2024) as an active TTA baseline, adapting it to use binary-feedback data by incorporating a complementary loss for negative samples. The non-active TTA and active TTA method accuracies are reported in Appendix C for comparison.

**Dataset.** To evaluate the robustness of BATTA-RL across various domain shifts, we utilized standard image corruption datasets CIFAR10-C, CIFAR100-C, and Tiny-ImageNet-C (Hendrycks & Dietterich, 2019). Additionally, we conducted experiments on the PACS dataset (Li et al., 2017), which is commonly used for domain adaptation tasks. For most experiments, we pre-trained the source model on the source domain while adapting and evaluating the model on the test-time domains. To more closely simulate real-world scenarios with evolving distribution shifts, we implemented a continual TTA setting (Wang et al., 2022) where corruption continuously changes.

**Settings and hyperparameters.** We configured BATTA-RL to operate with minimal labeling effort, using only 3 binary feedbacks within each 64-sample test batch, accounting for less than 5%. We utilize a single value of balancing hyperparameters $\alpha = 2$ and $\beta = 1$ for BATTA-RL in all experiments. A comprehensive list of settings and hyperparameters is provided in Appendix D.

**Overall result.** Table 1 presents the comprehensive results of our experiments on standard corruption benchmarks. BATTA-RL consistently and significantly outperformed all baseline methods across various corruption types and severity levels, demonstrating its effectiveness in binary-feedback active test-time adaptation scenarios. Notably, existing TTA methods, despite the use of binary-feedback data, exhibited suboptimal performance. This observation held even for advanced methods like EATA,

Table 1: Accuracy (%) comparisons with TTA and active TTA baselines with binary feedback in corruption datasets (severity level 5). Notation * indicates the modified algorithm to utilize binary-feedback samples. B: Binary-feedback active TTA. Results outperforming all other baselines are highlighted in **bold** fonts. Averaged over three random seeds. Comparison with non-active TTAs and full-label active TTA are in Table 9 in Appendix C.

| Label | Method | Noise | | | Blur | | | | Weather | | | | Digital | | | | |
|---|---|---|---|---|---|---|---|---|---|---|---|---|---|---|---|---|---|
| | | Gau. | Shot | Imp. | Def. | Gla. | Mot. | Zoom | Snow | Fro. | Fog | Brit. | Cont. | Elas. | Pix. | JPEG | Avg. |
| - | SrcValid | 25.97 | 33.19 | 24.71 | 56.73 | 52.02 | 67.37 | 64.80 | 77.97 | 67.01 | 74.14 | 91.51 | 33.90 | 76.62 | 46.38 | 73.23 | 57.23 |
| - | BN-Stats | 66.96 | 69.04 | 60.36 | 87.78 | 65.55 | 86.29 | 87.38 | 81.63 | 80.28 | 85.39 | 90.74 | 86.88 | 76.72 | 79.33 | 71.92 | 78.42 |
| B | TENT* | 75.11 | 79.68 | 70.89 | 82.24 | 67.17 | 77.85 | 82.43 | 79.48 | 80.43 | 80.50 | 86.64 | 84.81 | 72.83 | 78.18 | 71.88 | 78.01 |
| B | EATA* | 76.04 | 78.18 | 68.99 | 79.14 | 65.27 | 76.08 | 81.33 | 78.07 | 79.91 | 82.16 | 86.86 | 85.50 | 73.16 | 80.05 | 73.79 | 77.64 |
| B | SAR* | 71.27 | 78.41 | 72.68 | 88.92 | 72.62 | 88.00 | 89.63 | 86.18 | 86.64 | 87.61 | 92.39 | 90.07 | 81.55 | 86.44 | 80.43 | 83.52 |
| B | CoTTA* | 66.97 | 69.04 | 60.35 | 87.77 | 65.54 | 86.29 | 87.38 | 81.63 | 80.28 | 85.40 | 90.73 | 86.87 | 76.74 | 79.35 | 71.92 | 78.42 |
| B | RoTTA* | 67.06 | 71.87 | 64.74 | 82.99 | 69.58 | 85.91 | 89.60 | 85.09 | 87.08 | 87.44 | 91.74 | 87.76 | 81.29 | 82.35 | 81.41 | 81.06 |
| B | SoTTA* | 74.57 | 81.81 | 74.11 | 83.94 | 70.42 | 82.74 | 86.96 | 83.51 | 84.96 | 84.76 | 90.00 | 83.79 | 77.06 | 82.92 | 78.32 | 81.32 |
| B | SimATTA* | 48.21 | 65.38 | 57.69 | 68.10 | 63.19 | 75.74 | 83.06 | 80.10 | 82.40 | 83.26 | 88.75 | 75.73 | 77.30 | 78.39 | 79.23 | 73.77 |
| B | BATTA-RL | **76.78** | **84.24** | **78.75** | **87.51** | **77.39** | **88.38** | **91.36** | **89.42** | **90.72** | **90.30** | **94.65** | **92.62** | **86.15** | **92.42** | **87.24** | **87.20** |

(a) CIFAR10-C.

| Label | Method | Noise | | | Blur | | | | Weather | | | | Digital | | | | |
|---|---|---|---|---|---|---|---|---|---|---|---|---|---|---|---|---|---|
| | | Gau. | Shot | Imp. | Def. | Gla. | Mot. | Zoom | Snow | Fro. | Fog | Brit. | Cont. | Elas. | Pix. | JPEG | Avg. |
| - | SrcValid | 10.63 | 12.14 | 7.17 | 34.86 | 19.58 | 44.09 | 41.94 | 46.34 | 34.22 | 41.08 | 67.31 | 18.47 | 50.36 | 24.91 | 44.56 | 33.18 |
| - | BN-Stats | 39.23 | 40.75 | 34.10 | 66.14 | 42.46 | 63.57 | 64.82 | 53.81 | 53.49 | 58.15 | 68.22 | 64.48 | 53.88 | 56.63 | 45.17 | 53.66 |
| B | TENT* | 50.42 | 53.46 | 42.35 | 49.47 | 34.76 | 38.08 | 38.94 | 30.22 | 28.31 | 23.10 | 24.21 | 17.25 | 11.96 | 10.12 | 6.62 | 30.62 |
| B | EATA* | 13.31 | 5.29 | 4.98 | 4.46 | 3.89 | 3.96 | 3.86 | 3.68 | 3.47 | 3.36 | 3.76 | 2.78 | 3.24 | 3.30 | 3.51 | 4.46 |
| B | SAR* | 47.38 | 56.17 | 48.93 | 66.27 | 50.94 | 65.22 | **68.52** | 60.74 | 62.75 | 63.13 | 71.00 | **70.11** | 59.44 | 65.40 | 56.19 | 60.81 |
| B | CoTTA* | 39.24 | 40.75 | 34.10 | 66.13 | 42.48 | 63.57 | 64.83 | 53.80 | 53.46 | 58.16 | 68.22 | 64.47 | 53.89 | 56.66 | 45.16 | 53.66 |
| B | RoTTA* | 38.94 | 42.77 | 36.75 | 61.02 | 44.37 | 62.98 | 67.94 | 59.33 | 62.20 | 60.49 | 70.47 | 64.99 | 58.80 | 61.53 | 54.45 | 56.47 |
| B | SoTTA* | **52.10** | 57.66 | 48.67 | 61.16 | 48.45 | 62.72 | 67.51 | 59.40 | 61.53 | 62.96 | 69.49 | 67.00 | 56.91 | 62.84 | 56.58 | 59.67 |
| B | SimATTA* | 9.31 | 11.60 | 6.46 | 16.51 | 9.49 | 18.03 | 20.32 | 25.71 | 42.49 | 39.37 | 56.01 | 35.61 | 43.49 | 40.22 | 43.12 | 27.85 |
| B | BATTA-RL | 50.12 | **58.34** | **52.07** | **63.27** | **52.70** | **63.80** | 68.16 | **62.65** | **65.39** | **63.79** | **71.26** | 68.97 | **63.93** | **69.45** | **63.38** | **62.49** |

(b) CIFAR100-C.

| Label | Method | Noise | | | Blur | | | | Weather | | | | Digital | | | | |
|---|---|---|---|---|---|---|---|---|---|---|---|---|---|---|---|---|---|
| | | Gau. | Shot | Imp. | Def. | Gla. | Mot. | Zoom | Snow | Fro. | Fog | Brit. | Cont. | Elas. | Pix. | JPEG | Avg. |
| - | SrcValid | 6.99 | 8.93 | 5.09 | 15.18 | 9.65 | 26.50 | 26.33 | 29.77 | 33.64 | 12.34 | 31.80 | 2.34 | 27.71 | 34.99 | 46.97 | 21.22 |
| - | BN-Stats | 31.45 | 33.28 | 23.55 | 32.33 | 22.30 | 44.30 | 45.04 | 38.89 | 42.64 | 29.97 | 46.55 | 8.46 | 43.70 | 52.53 | 49.50 | 36.30 |
| B | TENT* | 35.56 | 34.58 | 20.65 | 13.74 | 5.05 | 4.84 | 3.46 | 2.62 | 2.01 | 1.98 | 1.93 | 1.35 | 1.64 | 1.72 | 1.61 | 8.85 |
| B | EATA* | 34.29 | 36.78 | 26.67 | 36.48 | 26.05 | 47.79 | 48.38 | 41.97 | 45.22 | 36.09 | 49.60 | 6.84 | 45.15 | 53.92 | 50.93 | 39.08 |
| B | SAR* | 33.60 | 38.47 | 29.34 | **35.46** | 27.41 | 47.15 | 48.48 | 41.28 | 45.48 | 36.93 | 50.47 | **13.47** | 46.37 | 52.99 | 50.76 | 39.85 |
| B | CoTTA* | 31.37 | 33.24 | 23.50 | 32.22 | 22.19 | 44.36 | 45.05 | 38.91 | 42.62 | 30.03 | 46.54 | 8.44 | 43.49 | 52.47 | 49.51 | 36.26 |
| B | RoTTA* | 31.84 | 34.96 | 25.67 | 30.91 | 25.07 | 45.53 | 47.12 | 41.51 | 44.79 | 31.41 | 47.21 | 12.90 | 43.82 | 49.07 | 48.83 | 37.38 |
| B | SoTTA* | **37.70** | **41.17** | **32.56** | 34.52 | **27.56** | 42.78 | 45.99 | 39.66 | 43.07 | 40.20 | 48.50 | 8.73 | 38.43 | 48.77 | 48.23 | 38.53 |
| B | SimATTA* | 14.40 | 24.46 | 15.14 | 14.63 | 13.34 | 30.87 | 35.56 | 25.23 | 34.33 | 19.95 | 34.33 | 1.62 | 34.47 | 43.55 | 45.49 | 25.82 |
| B | BATTA-RL | 33.16 | 37.75 | 28.21 | 34.97 | 26.27 | **48.57** | **49.42** | **43.11** | **47.16** | 37.84 | **51.41** | 10.01 | **47.21** | **54.03** | **52.72** | **40.12** |

(c) Tiny-ImageNet-C.

SAR, and SoTTA, which rely on fixed sample filtering strategies. Their reduced effectiveness in this setting highlights the limitations of such approaches when dealing with binary feedback and continuous distribution shifts. SimATTA, an active TTA baseline adapted for binary feedback, also struggled to maintain optimal performance. Its use of hard thresholding for sample selection, combined with incorrect-sample learning, likely contributed to noisy clustering and unstable adaptations.

We further examined the domain generalization capability in two scenarios: domain-wise data stream (continual TTA, Wang et al. (2022)) and mixed data stream (randomly mixed among all domains), following an existing study (Gui et al., 2024). In Table 2, BATTA-RL outperformed all baselines on average, demonstrating its ability to adapt effectively not only to corrupted images but also to broader domain shifts.

**Results on additional datasets.** We conduct an additional experiment to evaluate the scalability of BATTA-RL across various datasets covered in recent works (Lee et al., 2024a; Niu et al., 2023; Gui et al., 2024; Chen et al., 2022): ImageNet-C (Hendrycks & Dietterich, 2019), ImageNet-R (Hendrycks et al., 2021), ColoredMNIST (Arjovsky et al., 2019), VisDA-2021 (Bashkirova et al., 2022), and DomainNet (Peng et al., 2019).

Table 2: Accuracy (%) comparisons with TTA and active TTA baselines with binary feedback in PACS. The domain-wise data stream is a continual TTA setting, and the mixed data stream shuffled all domains randomly, where we report the cumulative accuracy at four adaptation points. Notation * indicates the modified algorithm to utilize binary-feedback samples. B: Binary-feedback active TTA. Results outperforming all other baselines are highlighted in **bold** fonts. Comparison with non-active TTAs and full-label active TTA are in Table 10 in Appendix C.

| Label | Method | Domain-wise data stream | | | | Mixed data stream | | | |
|---|---|---|---|---|---|---|---|---|---|
| | | Art | Cartoon | Sketch | Avg | 25% | 50% | 75% | 100%(Avg) |
| - | SrcValid | 59.38 ±0.00 | 27.94 ±0.21 | 42.96 ±0.01 | 43.43 ±0.07 | 42.74 ±1.13 | 42.80 ±0.22 | 42.64 ±0.30 | 42.77 ±0.01 |
| - | BN Stats | 67.87 ±0.18 | 63.48 ±0.88 | 54.07 ±0.36 | 61.81 ±0.18 | 59.09 ±0.29 | 58.28 ±0.08 | 58.05 ±0.22 | 57.82 ±0.20 |
| B | TENT* | 71.96 ±0.16 | 69.42 ±1.26 | 52.21 ±1.22 | 64.53 ±0.70 | **60.69 ±0.87** | 59.54 ±1.32 | 59.12 ±1.91 | 58.65 ±1.95 |
| B | EATA* | 68.75 ±0.26 | 65.32 ±0.65 | 58.86 ±0.89 | 64.31 ±0.23 | 59.71 ±0.13 | 59.64 ±0.46 | 60.08 ±0.63 | 60.43 ±0.29 |
| B | SAR* | 68.00 ±0.24 | 63.63 ±0.78 | 55.49 ±0.49 | 62.37 ±0.10 | 59.26 ±0.10 | 58.70 ±0.20 | 58.63 ±0.14 | 58.67 ±0.11 |
| B | CoTTA* | 67.87 ±0.18 | 63.47 ±0.90 | 54.07 ±0.36 | 61.80 ±0.19 | 59.10 ±0.32 | 58.29 ±0.09 | 58.06 ±0.23 | 57.83 ±0.22 |
| B | RoTTA* | 66.93 ±0.47 | 47.25 ±1.73 | 57.77 ±0.67 | 57.32 ±0.52 | 56.60 ±0.65 | 55.91 ±0.50 | 55.88 ±0.12 | 55.65 ±0.41 |
| B | SoTTA* | 70.05 ±0.95 | 38.93 ±1.26 | 30.58 ±4.41 | 46.52 ±1.25 | 53.61 ±3.14 | 53.68 ±3.57 | 54.80 ±2.93 | 55.54 ±2.48 |
| B | SimATTA* | 63.07 ±2.38 | 60.16 ±6.10 | 71.94 ±0.89 | 65.06 ±2.51 | 53.43 ±13.73 | 59.62 ±10.21 | 63.73 ±8.99 | 66.41 ±7.95 |
| B | BATTA-RL | **73.86 ±3.76** | **76.81 ±2.45** | **76.03 ±1.61** | **75.57 ±0.93** | 59.65 ±0.70 | **64.70 ±0.78** | **69.23 ±0.17** | **72.18 ±0.38** |

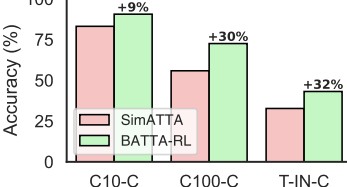

Figure 5: Accuracy (%) with full-labeled feedback (SimATTA) and binary-feedback (BATTA-RL) and under the equal total labeling cost. Averaged over three random seeds.

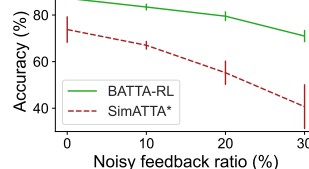

Figure 6: Accuracy (%) varying the feedback error in CIFAR10-C. Averaged over three random seeds.

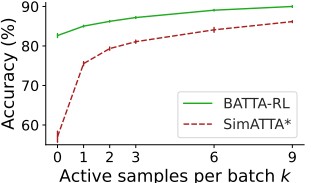

Figure 7: Accuracy (%) varying the number of active samples per batch ($k$) in CIFAR10-C. Averaged over three random seeds.

Results in Table 3 demonstrate a superior performance of BATTA-RL, especially on large-scale datasets such as ImageNet-C. The key insight is that BATTA-RL formulates both binary feedback and unlabeled sample adaptation as a single reinforcement learning objective, where the reward signals seamlessly guide the model's adaptation. Also, the use of MC-dropout provides a robust uncertainty estimate, while optimizing on MC-dropout prevents the TTA model from overfitting, therefore showing a stable adaptation in large-scale datasets.

**Comparison with active TTA.** To demonstrate the effectiveness of BATTA-RL, we compared it with the full-labeled active TTA method (SimATTA) under various datasets. We experimented with two scenarios: (1) an equal labeling cost (details in Appendix D.1, results in Figure 5) and (2) an equal number of active samples (Table 9 in Appendix C). We observe BATTA-RL is already outperforming SimATTA with an equal number of active samples. Moreover, we observe the **superior performance of BATTA-RL over full-label feedback active TTA (SimATTA)** when we provide more binary-feedback samples with an equal labeling cost with full-labeling. Our method is especially more effective with datasets that include a larger number of classes where full-label feedback is expensive. This showcases the importance of binary feedback's lightweight and effective nature compared to full-label feedback.

**Impact of labeling error.** We assumed the binary feedback provided by the oracle contained no labeling errors. In practice, user feedback might include labeling errors by shifting the binary feedback between *correct* and *incorrect*. We examine the impact of binary-feedback error compared to the full-label error in SimATTA. As shown in Figure 6, SimATTA shows significant accuracy degradation under labeling error by highly relying on the noisy labeled samples without utilizing unlabeled samples. In contrast, using many confident unlabeled samples could reduce the impact of labeling error; thereby, BATTA-RL consistently outperformed SimATTA and showed robustness by effectively utilizing both binary feedback and unlabeled samples.

Table 3: Accuracy (%) comparisons with TTA and active TTA baselines in additional datasets. Notation * indicates the modified algorithm to utilize binary-feedback samples. Results outperforming all other baselines are highlighted in **bold** fonts. Averaged over three random seeds.

| Dataset | SrcValid | BN-Stats | TENT* | EATA* | SAR* | CoTTA* | RoTTA* | SoTTA* | SimATTA* | BATTA-RL |
|---------|----------|----------|-------|-------|------|--------|--------|--------|----------|----------|
| ImageNet-C | 14.43 | 26.88 | 0.93 | 30.87 | 35.15 | 26.80 | 22.55 | 36.02 | 17.50 | **36.59** |
| ImageNet-R | 33.05 | 35.08 | 29.10 | 37.14 | 36.64 | 35.02 | 34.35 | 31.00 | 35.63 | **38.59** |
| VisDA-2021 | 27.36 | 26.46 | 20.38 | 27.82 | 27.41 | 26.46 | 27.23 | 27.71 | 22.80 | **29.30** |
| DomainNet | 54.82 | 54.41 | 18.80 | 59.49 | 57.78 | 54.40 | 56.41 | 54.82 | 58.41 | **60.85** |
| ColoredMNIST | 50.49 | 45.59 | 44.92 | 45.59 | 45.74 | 45.60 | 48.90 | 59.45 | 93.66 | **96.75** |

Table 4: Average wall-clock time per batch (s) comparisons with TTA and active TTA baselines with binary feedback in Tiny-ImageNet-C. Notation * indicates the modified algorithm to utilize binary-feedback samples. Averaged over three random seeds.

| | SrcValid | BN-Stats | TENT* | EATA* | SAR* | CoTTA* | RoTTA* | SoTTA* | SimATTA* | BATTA-RL |
|---|----------|----------|-------|-------|------|--------|--------|--------|----------|----------|
| Avg. | 0.18 ±0.12 | 0.33 ±0.20 | 1.03 ±0.35 | 0.98 ±0.39 | 1.02 ±0.38 | 26.63 ±5.40 | 1.68 ±0.27 | 1.25 ±0.16 | 45.45 ±13.50 | 4.19 ±0.06 |

**Effect of number of active samples.** We evaluated how the number of active samples per batch ($k$) influences adaptation performance. As illustrated in Figure 7, BATTA-RL maintains high accuracy even with a small number of active samples. The performance improves as $k$ increases, showcasing effective utilization of additional binary feedback. SimATTA shows a similar trend of increasing accuracy with more active samples, but the overall performance is consistently lower than BATTA-RL. This suggests that BATTA-RL can effectively leverage additional feedback, indicating its potential for deployment in scenarios with varying labeling budgets.

**Synergistic effect of adaptation strategies.** We compared BATTA-RL against its components: Binary Feedback-guided Adaptation (BFA) and Agreement-based self-Adaptation (ABA). In CIFAR-10-C, we observed that BFA-only adaptation achieved 58.90% and ABA-only adaptation achieved 82.64%, where BATTA-RL achieved on average 87.20% accuracy, consistently outperforming entire continual corruptions. The superior performance of the combined approach (BATTA-RL) demonstrates that BFA and ABA complement each other to achieve robust accuracy.

**Runtime analysis.** To assess the practical applicability of BATTA-RL, we conducted a comprehensive runtime analysis by measuring the average wall-clock time per batch across different methods on the Tiny-ImageNet-C dataset. Our results in Table 4 show that BATTA-RL requires 4.19 ±0.06 seconds per batch, positioning it between simpler TTA methods (0.33-1.68s) and more complex approaches like CoTTA (26.63s) and SimATTA (45.45s). The runtime profile demonstrates that BATTA-RL achieves a favorable balance between computational cost and performance, particularly considering its significant accuracy improvements over faster baselines while maintaining substantially lower processing time than methods like SimATTA.

## 5 RELATED WORK

**Test-time adaptation.** Test-time adaptation (TTA) improves model accuracy on distribution shift on the pre-trained model with only unlabeled test samples (Wang et al., 2021). Existing TTA focused on robust adaptation (Niu et al., 2023; Gong et al., 2022; Yuan et al., 2023; Wang et al., 2022; Boudiaf et al., 2022; Niu et al., 2022; Gong et al., 2023b; Park et al., 2024) across various types of distribution shifts (Niu et al., 2023; Gong et al., 2022; Wang et al., 2022; Gong et al., 2023b; Press et al., 2023). However, existing TTA methods suffer from adaptation failures during lifelong adaptation (Press et al., 2023), stressing the need for a few-sample guide for robust adaptation. Active test-time adaptation (ATTA) (Gui et al., 2024) introduced a foundational analysis of active TTA setting. It proposed a supervised learning scheme (SimATTA) using low-entropy source-like sample pseudo-labeling and active labeling from an incremental clustering algorithm. However, SimATTA is sensitive to the pre-trained model and selected active samples, as it does not leverage most unlabeled samples and only utilizes a few labeled samples. In contrast, BATTA utilizes a large set of unlabeled samples while guiding adaptation with binary-feedback samples, performing more stable than SimATTA.

**Active learning.** Active learning (Cohn et al., 1994; Settles, 2009) involves an oracle (e.g., human annotator) in the machine learning process to develop efficient annotation and training procedures. Active learning framework has been widely studied in active (source-free) domain adaptation (Ash et al., 2019; Prabhu et al., 2021; Li et al., 2022; Wang et al., 2023; Du & Li, 2024; Kothandaraman et al., 2023; Ning et al., 2021) and active TTA (Gui et al., 2024). Compared with active domain adaptation, active TTA focuses on the non-regrettable active sample selection on the continuously changing data stream without access to source data. Using binary feedback is related to the active learning with partial feedback (ALPF) problem (Hu et al., 2019), which seeks to recursively obtain partial labels until a definitive label is identified. Joshi et al. (2010) proposed a binary feedback active learning setup where users compare two images and report whether they belong to the same category. In contrast, our approach leverages single-step binary feedback on the model's current batch sample output without requiring additional data. This simplifies the process and reduces the labeling effort.

**Reinforcement learning for model tuning.** Reinforcement learning (RL) has been successfully applied in various domains to incorporate non-differentiable rewards in the optimization process (Zoph & Le, 2017; Yoon et al., 2020; Ouyang et al., 2022; Fan et al., 2023; Black et al., 2024). For example, Zoph & Le (2017) and Yoon et al. (2020) employ the REINFORCE algorithm to use the accuracy of the validation dataset as a (non-differentiable) reward in neural architecture search or data valuation. In the domain of the natural language process, reinforcement learning with human feedback (RLHF) (Ouyang et al., 2022) has gained prominence for fine-tuning large language models. Such a recipe has been extended to the domain of computer vision such as fine-tuning text-to-image diffusion models using human feedback (Fan et al., 2023; Black et al., 2024). Similar approaches have been explored in vision and multi-modal research (Le et al., 2022; Pinto et al., 2023). Recently, Reinforcement Learning with CLIP Feedback (RLCF, Zhao et al. (2023)) has been proposed for test-time adaptation of vision-language models. RLCF relies on the pre-trained CLIP model as a reward function, which may not be available or suitable for all domains or tasks. In contrast, our approach provides a more general and flexible approach for test-time adaptation by effectively guiding the adaptation without relying on specific pre-trained models.

## 6 CONCLUSION

We proposed binary-feedback active test-time adaptation (BATTA) to address the challenge of adapting pre-trained models to new domains with minimal labeling effort. Our approach leverages binary feedback on the model predictions (*correct* or *incorrect*) from an oracle to guide the adaptation process, significantly reducing the labeling cost than existing methods requiring full-class labels. Our method, BATTA-RL, uniquely combines binary feedback-guided adaptation on uncertain samples with agreement-based self-adaptation on confident samples in a reinforcement learning framework, balancing between a few labeled samples and many unlabeled samples. Through extensive experiments on distribution shift datasets, we demonstrated that BATTA-RL outperforms state-of-the-art test-time adaptation methods, showcasing its effectiveness in handling continuous distribution shifts. Overall, BATTA represents a significant step forward in test-time adaptation, offering a practical balance between performance and labeling efficiency.

## REPRODUCIBILITY STATEMENT

We provide the source code in the supplementary material with the instructions to prepare the dataset. We specify the experimental details in Appendix D, including datasets, scenarios, and hyperparameters.

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

# Appendix

## Binary-Feedback Active Test-Time Adaptation

## A DISCUSSION

### A.1 LIMITATIONS AND FUTURE WORKS

Despite the promising results, BATTA and BATTA-RL have limitations. First, the reliance on binary feedback, while reducing labeling effort, may still require substantial oracle involvement in scenarios with high data variability or rapid domain shifts. Future work will explore reducing oracle involvement by developing more advanced and dynamic sample selection strategies. Second, the computational overhead introduced by Monte Carlo dropout, although manageable, could be significant in resource-constrained environments. This overhead could be reduced by efficient TTA (Hong et al., 2023; Song et al., 2023) and on-device machine learning (Liberis & Lane, 2023; Rusci et al., 2020; Gong et al., 2023a). Finally, our method assumes that the oracle feedback is always accurate, which might not hold in some real-world scenarios. Although our algorithm showed robustness compared to the baseline, designing a method for handling noisy or incorrect feedback remains an area for future research.

### A.2 SOCIETAL IMPACTS

#### A.2.1 POSITIVE IMPACTS

- **Reduced labeling costs.** For Binary-feedback Active TTA (BATTA), the use of binary feedback significantly reduces the need for extensive labeling, lowering costs and making advanced machine learning techniques more accessible to smaller organizations and underfunded research projects.

- **Improved adaptability in real-world applications.** By enabling models to adapt in real-time with minimal labeling, BATTA can enhance the performance of applications like autonomous driving, healthcare diagnostics, and personalized recommendations, improving safety, efficiency, and user experience.

- **Enhanced robustness and accuracy.** BATTA-RL's robust adaptation mechanism can improve the accuracy of models in diverse and changing environments, leading to more reliable and trustworthy AI systems in critical applications such as medical imaging and environmental monitoring.

#### A.2.2 NEGATIVE IMPACTS

- **Dependence on oracle feedback.** The reliance on binary feedback from oracles could still be a bottleneck in some applications, particularly if the feedback is not accurate or timely, potentially limiting the method's effectiveness in highly dynamic environments.

- **Potential for misuse.** Like any advanced AI technology, BATTA-RL could be misused in applications that require constant adaptation to new data, such as surveillance or targeted advertising, potentially leading to privacy concerns or biased decision-making.

- **Computational overhead.** The use of Monte Carlo dropout and other advanced techniques might increase computational requirements, potentially limiting the method's applicability in resource-constrained environments or contributing to higher energy consumption. Recent advances in efficient TTA (Hong et al., 2023; Song et al., 2023) and on-device learning (Liberis & Lane, 2023; Rusci et al., 2020; Gong et al., 2023a) could be integrated to reduce the computational overhead and enhance the applicability in resource-constrained environments.

## B ADDITIONAL STUDIES

**Impact of prediction agreement.** To assess the effectiveness of our prediction agreement method for confident sample selection, we compared it against fixed confidence thresholding approaches. We evaluated thresholds ranging from 0.8 to 0.99, with 0.99 being the value used in SoTTA (Gong et al., 2023b). Figure 8 illustrates the performance of these approaches on unlabeled-only TTA in

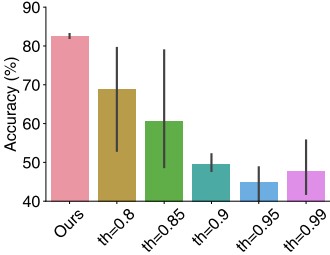 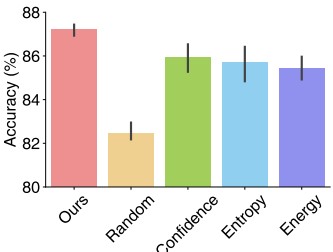 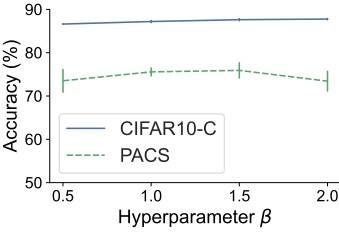

Figure 8: Accuracy (%) by confidence thresholding strategies: Ours and hard thresholding (th). Averaged over three random seeds.

Figure 9: Accuracy (%) varying the binary-feedback active sample selection strategy in CIFAR10-C. Averaged over three random seeds.

Figure 10: Accuracy (%) varying the balancing hyperparameter ($\beta$) in CIFAR10-C and PACS. Averaged over three random seeds.

Table 5: Accuracy (%) comparisons with varying epochs in CIFAR10-C (severity level 5). B: Binary-feedback active TTA. Averaged over three random seeds.

| Label | Method | Noise | | | Blur | | | | Weather | | | | Digital | | | | |
|-------|--------|-------|------|------|------|------|------|------|------|------|------|------|------|------|------|------|------|
| | | Gau. | Shot | Imp. | Def. | Gla. | Mot. | Zoom | Snow | Fro. | Fog | Brit. | Cont. | Elas. | Pix. | JPEG | Avg. |
| B | BATTA-RL (epoch = 3) | 76.78 | 84.24 | 78.75 | 87.51 | 77.39 | 88.38 | 91.36 | 89.42 | 90.72 | 90.30 | 94.65 | 92.62 | 86.15 | 92.42 | 87.24 | **87.20** |
| B | · epoch = 1 | 76.92 | 84.29 | 78.61 | 86.99 | 77.20 | 88.36 | 91.51 | 89.31 | 90.58 | 90.30 | 94.51 | 92.70 | 85.77 | 92.08 | 87.50 | 87.11 |
| B | · epoch = 2 | 76.30 | 84.01 | 78.80 | 87.66 | 77.30 | 88.43 | 91.56 | 89.16 | 90.61 | 90.37 | 94.52 | 92.61 | 85.83 | 92.33 | 87.75 | 87.15 |

the continual CIFAR10-C setting. Our prediction agreement method consistently outperformed all fixed thresholding approaches, which exhibited high variance and instability. This result demonstrates the superiority of our dynamic sample selection strategy, particularly in scenarios with continuously changing corruptions, highlighting the importance of adaptive confidence assessment in test-time adaptation.

**Impact of sample selection.** We examined the impact of sample selection, including our MC-dropout certainty approach with random selection, maximum entropy (Saito et al., 2020), minimum confidence (Sohn et al., 2020), and minimum energy (Liu et al., 2020). In Figure 9, our method outperforms others by leveraging MC-dropout to estimate epistemic uncertainty. In contrast, naive methods may struggle with overconfidence in test-time scenarios, failing to prioritize samples that offer the most valuable information for model improvement.

**Sensitivity to balancing hyperparameter $\alpha, \beta$** . We investigated the sensitivity of BATTA-RL to the balancing hyperparameter $\beta$ while fixing $\alpha = 2.0$, which controls the trade-off between binary feedback-guided adaptation and agreement-based self-adaptation. Figure 10 illustrates the overall accuracy across various $\beta$ values for both image corruption and domain adaptation datasets. The results demonstrate that BATTA-RL maintains consistent performance across a wide range of $\beta$ values, indicating robustness to this hyperparameter choice. This stability suggests that BATTA-RL can effectively deploy across different scenarios without extensive hyperparameter tuning.

**Impact of the number of epochs.** To understand the BATTA-RL's performance under time-constrained environments, we examined how reducing training epochs affects adaptation accuracy on CIFAR10-C. We compared our standard 3-epoch configuration against reduced 1- and 2-epoch settings, adjusting learning rates proportionally (×3 and ×1.5) to compensate for fewer update steps. Results in Table 5 show that BATTA-RL maintains robust performance even with fewer epochs. This consistent performance across epoch configurations demonstrates that BATTA-RL can effectively adapt to distribution shifts even under stricter computational constraints, offering flexibility in real-world deployment scenarios where faster adaptation may be preferred.

Table 6: Accuracy (%) comparisons with augmentation-based uncertainty estimation in CIFAR10-C (severity level 5). B: Binary-feedback active TTA. Averaged over three random seeds.

| Label | Method | Noise | | | Blur | | | | Weather | | | | Digital | | | | |
|---|---|---|---|---|---|---|---|---|---|---|---|---|---|---|---|---|---|
| | | Gau. | Shot | Imp. | Def. | Gla. | Mot. | Zoom | Snow | Fro. | Fog | Brit. | Cont. | Elas. | Pix. | JPEG | Avg. |
| B | BATTA-RL | 76.78 | 84.24 | 78.75 | 87.51 | 77.39 | 88.38 | 91.36 | 89.42 | 90.72 | 90.30 | 94.65 | 92.62 | 86.15 | 92.42 | 87.24 | **87.20** |
| B | · Augmentation | 66.22 | 46.99 | 25.43 | 18.49 | 12.82 | 11.96 | 11.68 | 11.43 | 12.24 | 11.37 | 11.48 | 10.87 | 11.45 | 11.96 | 11.71 | 19.07 |

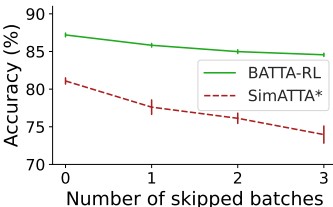

Figure 11: Accuracy (%) varying the labeling skip in CIFAR10-C. Averaged over three random seeds.

**Replacing MC-dropout with augmentation.** To further understand the impact of uncertainty estimation, we compare BATTA-RL with replacing MC-dropout with augmentation-based uncertainty estimation Wang et al. (2022); Zhang et al. (2022). Results in Table 6 suggest that augmentation-based uncertainty appears less stable and overfits in the early adaptation stage, leading to suboptimal performance.

**Impact of intermittent labeling.** To further understand the impact of the annotator's labeling budget, we conduct an experiment scenario where annotators skip labeling a few batches (e.g., labeling only 1 out of 4 consecutive batches). In Figure 11, we observe our BATTA-RL shows stable performance with minimal degradation, whereas the active TTA baseline (SimATTA) shows high accuracy degradation with batch skips.

## C  ADDITIONAL RESULTS

**Results on additional scenarios.** Recent TTA works suggest a new scenario of (1) imbalanced/non-iid label distribution, where ground-truth labels are temporally correlated (Niu et al., 2023; Gong et al., 2022), (2) and batch size 1 (Niu et al., 2023). Note that SimATTA's clustering algorithm for sample selection is not applicable in scenarios where the memory capacity is limited to only one image. Experiment results on CIFAR10-C (Table 7) suggest the robustness of our method over imbalanced label distribution and batch size 1 by effectively utilizing reward signals from the binary feedback and unlabeled samples.

**Results on ResNet50.** To further examine the applicability of BATTA-RL in various model architectures, we experimented with ResNet50. Table 8 shows the overall result, where BATTA-RL still outperformed the baselines in all corruptions. The result demonstrates the feasibility of BATTA-RL.

**Comparison with original TTA and active TTA.** In Table 9 and 10, we compare BATTA-RL with **original TTA (without binary-feedback samples) and original active TTA (with full-labeling)** baselines. Experiment results demonstrate the superior performance of BATTA-RL, even outperforming the active TTA baseline (SimATTA, Gui et al. (2024)), showing the effectiveness of our RL-based adaptation with binary-feedback adaptation and agreement-based adaptation. We consider this the drawback of SimATTA's strategy of using source-like confident samples. Even with tuning the

Table 7: Accuracy (%) comparisons with TTA and active TTA baselines with binary feedback in online CIFAR10-C (severity level 5) with additional scenarios. Notation * indicates the modified algorithm to utilize binary-feedback samples. B: Binary-feedback active TTA. Results outperforming all other baselines are highlighted in **bold** fonts. Averaged over three random seeds.

| Label | Method | Noise | | | Blur | | | | Weather | | | | Digital | | | | |
|---|---|---|---|---|---|---|---|---|---|---|---|---|---|---|---|---|---|
| | | Gau. | Shot | Imp. | Def. | Gla. | Mot. | Zoom | Snow | Fro. | Fog | Brit. | Cont. | Elas. | Pix. | JPEG | Avg. |
| - | SrcValid | 24.01 | 30.91 | 22.36 | 55.00 | 53.44 | 66.99 | 63.74 | 78.01 | 68.41 | 73.92 | 91.34 | 34.30 | 76.77 | 46.26 | 73.05 | 57.70 |
| - | BN-Stats | 22.75 | 23.33 | 20.83 | 30.15 | 21.45 | 29.38 | 28.90 | 27.33 | 28.05 | 29.27 | 31.37 | 31.06 | 25.21 | 26.37 | 22.91 | 26.58 |
| B | TENT* | 20.00 | 21.27 | 19.56 | 26.77 | 19.19 | 26.54 | 25.76 | 24.94 | 24.66 | 26.50 | 28.03 | 26.66 | 22.14 | 23.88 | 20.98 | 23.79 |
| B | EATA* | 16.24 | 16.52 | 13.73 | 18.82 | 15.97 | 18.87 | 18.79 | 16.87 | 17.62 | 19.30 | 20.34 | 17.85 | 18.02 | 17.87 | 16.29 | 17.54 |
| B | SAR* | 22.95 | 23.57 | 21.36 | 30.06 | 21.44 | 29.52 | 28.81 | 27.38 | 28.10 | 29.48 | 31.40 | 30.69 | 24.90 | 26.46 | 23.31 | 26.63 |
| B | CoTTA* | 22.76 | 23.36 | 21.14 | 29.99 | 21.42 | 29.48 | 28.84 | 27.42 | 28.10 | 29.43 | 31.34 | 30.85 | 24.92 | 26.43 | 23.22 | 26.58 |
| B | RoTTA* | 41.83 | 44.60 | 37.97 | 58.54 | 41.14 | 57.40 | 57.79 | 52.54 | 51.86 | 56.87 | 62.27 | 53.20 | 48.41 | 50.65 | 44.84 | 50.66 |
| B | SoTTA* | 67.03 | 71.31 | 61.84 | 83.96 | 66.01 | 82.23 | 84.47 | 78.62 | 78.48 | 82.94 | 87.74 | 77.29 | 74.07 | 76.94 | 72.12 | 76.34 |
| B | SimATTA* | 59.05 | 68.67 | 44.43 | 84.96 | 67.46 | 83.36 | 84.99 | 81.75 | 82.87 | 83.83 | 89.11 | 72.28 | 76.15 | 81.90 | 73.41 | 75.62 |
| B | **BATTA-RL** | **82.32** | **84.02** | **75.77** | **90.39** | **79.05** | **90.73** | **90.93** | **90.71** | **89.09** | **92.22** | **95.36** | **82.16** | **87.56** | **87.40** | **85.91** | **86.91** |

(a) Imbalanced (non-iid) label distribution.

| Label | Method | Noise | | | Blur | | | | Weather | | | | Digital | | | | |
|---|---|---|---|---|---|---|---|---|---|---|---|---|---|---|---|---|---|
| | | Gau. | Shot | Imp. | Def. | Gla. | Mot. | Zoom | Snow | Fro. | Fog | Brit. | Cont. | Elas. | Pix. | JPEG | Avg. |
| - | SrcValid | 25.96 | 33.19 | 24.71 | 56.73 | 52.02 | 67.37 | 64.80 | **77.97** | 67.00 | 74.14 | **91.50** | 33.90 | 76.61 | 46.38 | **73.23** | 57.70 |
| - | BN-Stats | 20.53 | 21.09 | 18.15 | 32.45 | 20.72 | 33.45 | 30.49 | 28.76 | 29.29 | 33.34 | 36.96 | 40.55 | 24.20 | 25.95 | 21.43 | 27.82 |
| B | TENT* | 10.50 | 10.01 | 10.01 | 10.01 | 10.01 | 10.01 | 10.01 | 10.01 | 10.01 | 10.01 | 10.01 | 10.01 | 10.01 | 10.01 | 10.01 | 10.04 |
| B | EATA* | 20.53 | 21.09 | 18.15 | 32.45 | 20.72 | 33.45 | 30.49 | 28.76 | 29.29 | 33.34 | 36.96 | 40.55 | 24.20 | 25.95 | 21.43 | 27.82 |
| B | SAR* | 20.56 | 21.12 | 18.29 | 32.51 | 20.86 | 33.59 | 30.67 | 29.12 | 29.51 | 33.68 | 37.52 | 41.15 | 24.70 | 26.57 | 21.98 | 28.12 |
| B | CoTTA* | 20.54 | 21.09 | 18.15 | 32.44 | 20.70 | 33.45 | 30.49 | 28.75 | 29.28 | 33.33 | 36.95 | 40.55 | 24.20 | 25.95 | 21.42 | 27.82 |
| B | RoTTA* | 11.70 | 10.23 | 10.03 | 10.01 | 10.01 | 10.01 | 10.01 | 10.01 | 10.01 | 10.01 | 10.01 | 10.01 | 10.01 | 10.01 | 10.01 | 10.14 |
| B | SoTTA* | 17.02 | 15.32 | 13.00 | 79.00 | 18.17 | 57.44 | 63.39 | 51.26 | 49.67 | 61.47 | 64.84 | 50.27 | 53.56 | 42.18 | 52.14 | 45.92 |
| B | **BATTA-RL** | **62.14** | **64.01** | **55.13** | **82.07** | **59.64** | **79.22** | **83.26** | 75.84 | **71.26** | **81.92** | 86.13 | 31.94 | 71.34 | **73.80** | 67.73 | **70.17** |

(b) Batch size 1.

Table 8: Accuracy (%) comparisons with TTA and active TTA baselines with binary feedback in CIFAR10-C (severity level 5) with ResNet50. Notation * indicates the modified algorithm to utilize binary-feedback samples. B: Binary-feedback active TTA. Results outperforming all other baselines are highlighted in **bold** fonts. Averaged over three random seeds.

| Label | Method | Noise | | | Blur | | | | Weather | | | | Digital | | | | |
|---|---|---|---|---|---|---|---|---|---|---|---|---|---|---|---|---|---|
| | | Gau. | Shot | Imp. | Def. | Gla. | Mot. | Zoom | Snow | Fro. | Fog | Brit. | Cont. | Elas. | Pix. | JPEG | Avg. |
| - | SrcValid | 22.56 | 27.66 | 21.49 | 46.91 | 43.23 | 55.29 | 54.62 | 66.90 | 53.91 | 61.31 | 84.94 | 24.24 | 65.29 | 41.03 | 65.35 | 48.98 |
| - | BN-Stats | 60.20 | 62.13 | 55.50 | 82.21 | 58.39 | 80.01 | 81.65 | 75.67 | 73.78 | 78.92 | 86.14 | 81.86 | 69.56 | 73.34 | 67.23 | 72.44 |
| B | TENT* | 67.91 | 72.96 | 63.60 | 72.68 | 56.98 | 62.43 | 65.48 | 60.95 | 58.81 | 56.47 | 66.26 | 64.45 | 55.80 | 61.30 | 57.70 | 61.58 |
| B | EATA* | 75.19 | 80.89 | 73.29 | 81.65 | 67.68 | 76.30 | 79.09 | 75.80 | 77.09 | 76.19 | 82.23 | 79.64 | 68.67 | 74.07 | 70.10 | 75.62 |
| B | SAR* | 63.51 | 70.85 | 65.95 | 85.07 | 66.46 | 84.06 | 86.33 | 82.68 | 83.24 | 84.02 | 90.46 | 86.74 | 78.53 | 83.68 | 79.10 | 79.53 |
| B | CoTTA* | 60.20 | 62.13 | 55.50 | 82.20 | 58.40 | 80.01 | 81.65 | 75.68 | 73.78 | 78.92 | 86.14 | 81.87 | 69.55 | 73.36 | 67.20 | 72.63 |
| B | RoTTA* | 60.77 | 65.94 | 60.67 | 79.87 | 65.04 | 82.22 | 86.19 | 82.03 | 84.23 | 84.58 | 90.00 | 85.51 | 79.68 | 81.57 | 81.19 | 77.88 |
| B | SoTTA* | 71.06 | 80.72 | 73.98 | 82.02 | 67.78 | 79.96 | 83.85 | 81.16 | 81.96 | 80.95 | 87.10 | 82.77 | 74.12 | 78.02 | 75.80 | 78.29 |
| B | SimATTA* | 33.37 | 49.99 | 41.33 | 62.69 | 58.03 | 76.02 | 81.32 | 77.35 | 80.75 | 79.95 | 88.83 | 67.17 | 76.13 | 72.59 | 78.84 | 68.29 |
| B | **BATTA-RL** | **75.72** | **83.25** | **78.58** | **85.41** | **75.75** | **86.14** | **89.82** | **87.28** | **89.55** | **88.83** | **93.67** | **92.04** | **84.93** | **91.91** | **88.38** | **86.08** |

hyperparameters, the accuracy of source-like samples is highly dependent on the source-pretrained model. This results in noisy predictions, hindering its applicability in various datasets and scenarios.

**Comparison with enhanced TTA.** Following the setting of SimATTA (Gui et al., 2024), we compare BATTA-RL with an enhanced TTA setting, which is **unsupervised TTA baselines adapting on the fine-tuned model**, which is tuned with an equal amount of binary-feedback active samples before the adaptation phase. In Table 11, we observe that BATTA-RL still outperforms over enhanced TTA baselines. The result necessitates the superiority of online adaptation on binary feedback samples.

Table 9: Accuracy (%) and standard deviation comparisons with **original TTA and full-label active TTA baselines** in corruption datasets (severity level 5). F: Full-label feedback active TTA, B: Binary-feedback active TTA. Results that outperform all baselines are highlighted in **bold** font. Averaged over three random seeds.

| Label | Method | Noise | | | Blur | | | | Weather | | | | Digital | | | | Avg. |
|---|---|---|---|---|---|---|---|---|---|---|---|---|---|---|---|---|---|
| | | Gau. | Shot | Imp. | Def. | Gla. | Mot. | Zoom | Snow | Fro. | Fog | Brit. | Cont. | Elas. | Pix. | JPEG | |
| - | SrcValid | 25.97 | 33.19 | 24.71 | 56.73 | 52.02 | 67.37 | 64.80 | 77.97 | 67.01 | 74.14 | 91.51 | 33.90 | 76.62 | 46.38 | 73.23 | 57.23 |
| - | BN-Stats | 66.96 | 69.04 | 60.36 | 87.78 | 65.55 | 86.29 | 87.38 | 81.63 | 80.28 | 85.39 | 90.74 | 86.88 | 76.72 | 79.33 | 71.92 | 78.42 |
| - | TENT | 74.34 | 77.30 | 65.86 | 74.12 | 54.40 | 58.08 | 58.89 | 53.49 | 50.45 | 46.76 | 48.23 | 40.65 | 34.78 | 34.37 | 29.62 | 53.42 |
| - | EATA | 76.45 | 77.33 | 64.70 | 77.51 | 62.31 | 71.91 | 78.34 | 75.29 | 75.24 | 78.56 | 84.68 | 83.19 | 68.81 | 70.97 | 67.18 | 74.16 |
| - | SAR | 67.94 | 69.45 | 62.82 | 87.79 | 66.18 | 86.31 | 87.38 | 81.63 | 80.28 | 85.39 | 90.74 | 86.88 | 76.72 | 79.33 | 71.98 | 78.72 |
| - | CoTTA | 66.97 | 69.04 | 60.37 | 87.78 | 65.55 | 86.30 | 87.38 | 81.63 | 80.27 | 85.39 | 90.74 | 86.88 | 76.72 | 79.33 | 71.92 | 78.42 |
| - | RoTTA | 65.21 | 71.11 | 64.77 | 85.11 | 69.73 | 87.44 | 89.95 | 86.05 | 86.60 | 87.98 | 92.73 | 88.00 | 82.53 | 85.49 | 81.11 | 81.59 |
| - | SoTTA | 74.59 | 81.22 | 74.55 | 84.74 | 71.41 | 83.33 | 87.86 | 83.68 | 84.63 | 85.51 | 90.34 | 83.09 | 78.87 | 82.88 | 77.99 | 81.65 |
| F | SimATTA | 73.89 | 82.45 | 73.36 | 79.97 | 72.14 | 84.13 | 88.95 | 86.22 | 89.01 | 87.94 | 92.81 | 85.21 | 80.94 | 85.93 | 83.97 | 83.13 |
| B | BATTA-RL | **76.78** | **84.24** | **78.75** | **87.51** | **77.39** | **88.38** | **91.36** | **89.42** | **90.72** | **90.30** | **94.65** | **92.62** | **86.15** | **92.42** | **87.24** | **87.20** |

(a) CIFAR10-C.

| Label | Method | Noise | | | Blur | | | | Weather | | | | Digital | | | | Avg. |
|---|---|---|---|---|---|---|---|---|---|---|---|---|---|---|---|---|---|
| | | Gau. | Shot | Imp. | Def. | Gla. | Mot. | Zoom | Snow | Fro. | Fog | Brit. | Cont. | Elas. | Pix. | JPEG | |
| - | SrcValid | 10.63 | 12.14 | 7.17 | 34.86 | 19.58 | 44.09 | 41.94 | 46.34 | 34.22 | 41.08 | 67.31 | 18.47 | 50.36 | 24.91 | 44.56 | 33.18 |
| - | BN-Stats | 39.23 | 40.75 | 34.10 | 66.14 | 42.46 | 63.57 | 64.82 | 53.81 | 53.49 | 58.15 | 68.22 | 64.48 | 53.88 | 56.63 | 45.17 | 53.66 |
| - | TENT | 49.71 | 51.12 | 38.34 | 42.40 | 24.86 | 21.51 | 17.21 | 9.39 | 5.84 | 4.24 | 3.87 | 2.56 | 2.74 | 2.40 | 2.36 | 18.57 |
| - | EATA | 10.40 | 2.88 | 2.81 | 2.50 | 2.22 | 2.21 | 1.99 | 2.17 | 1.91 | 1.65 | 1.53 | 1.23 | 1.25 | 1.12 | 1.05 | 2.46 |
| - | SAR | 46.45 | 55.24 | 48.53 | 66.27 | 50.93 | **65.35** | **68.49** | 60.73 | 62.36 | 63.37 | 71.12 | 69.48 | 59.76 | 65.34 | 56.33 | 60.65 |
| - | CoTTA | 39.24 | 40.75 | 34.11 | 66.13 | 42.46 | 63.57 | 64.82 | 53.81 | 53.49 | 58.14 | 68.22 | 64.48 | 53.87 | 56.63 | 45.17 | 53.66 |
| - | RoTTA | 35.63 | 40.04 | 35.55 | 60.32 | 42.09 | 62.76 | 67.53 | 58.54 | 60.60 | 60.72 | **71.58** | 64.08 | 59.50 | 63.13 | 54.49 | 55.77 |
| - | SoTTA | **52.31** | 57.80 | 48.30 | 61.57 | 48.82 | 63.45 | 68.17 | 59.54 | 61.69 | 62.62 | 69.73 | 66.30 | 57.40 | 63.35 | 56.67 | 59.85 |
| F | SimATTA | 42.86 | 54.18 | 44.18 | 53.98 | 46.64 | 60.51 | 65.54 | 57.01 | 62.73 | 57.25 | 68.38 | 52.17 | 54.53 | 61.10 | 56.88 | 55.86 |
| B | BATTA-RL | 50.12 | **58.34** | **52.07** | **63.27** | **52.70** | 63.80 | 68.16 | **62.65** | **65.39** | **63.79** | 71.26 | **68.97** | **63.93** | **69.45** | **63.38** | **62.49** |

(b) CIFAR100-C.

| Label | Method | Noise | | | Blur | | | | Weather | | | | Digital | | | | Avg. |
|---|---|---|---|---|---|---|---|---|---|---|---|---|---|---|---|---|---|
| | | Gau. | Shot | Imp. | Def. | Gla. | Mot. | Zoom | Snow | Fro. | Fog | Brit. | Cont. | Elas. | Pix. | JPEG | |
| - | SrcValid | 6.99 | 8.93 | 5.09 | 15.18 | 9.65 | 26.50 | 26.33 | 29.77 | 33.64 | 12.34 | 31.80 | 2.34 | 27.71 | 34.99 | 46.97 | 21.22 |
| - | BN-Stats | 31.45 | 33.28 | 23.55 | 32.33 | 22.30 | 44.30 | 45.04 | 38.89 | 42.64 | 29.97 | 46.55 | 8.46 | 43.70 | 52.53 | 49.50 | 36.30 |
| - | TENT | 35.97 | 33.92 | 18.12 | 8.67 | 2.93 | 2.84 | 2.57 | 2.35 | 1.87 | 1.86 | 1.86 | 1.33 | 1.57 | 1.63 | 1.58 | 7.94 |
| - | EATA | 34.53 | 36.80 | 26.46 | **36.49** | 25.69 | 47.83 | 48.33 | 41.88 | 44.98 | 35.83 | 49.62 | 6.86 | 44.86 | 53.79 | 50.95 | 38.99 |
| - | SAR | 33.35 | 38.03 | 28.94 | 35.83 | **27.12** | 47.13 | 48.39 | 41.36 | 45.09 | 36.79 | 50.24 | **13.46** | 46.45 | 52.44 | 50.52 | 39.68 |
| - | CoTTA | 31.45 | 33.29 | 23.54 | 32.35 | 22.27 | 44.33 | 44.99 | 38.94 | 42.67 | 29.99 | 46.57 | 8.67 | 43.74 | 52.58 | 49.45 | 36.32 |
| - | RoTTA | 31.13 | 34.94 | 25.71 | 31.74 | 25.01 | 46.18 | 47.47 | 41.40 | 45.13 | 31.38 | 48.01 | 8.92 | 45.07 | 50.77 | 49.69 | 37.50 |
| - | SoTTA | **37.62** | **40.91** | **31.72** | 33.55 | 26.75 | 41.50 | 44.84 | 37.72 | 41.42 | 38.57 | 47.04 | 7.46 | 34.88 | 44.08 | 45.04 | 36.89 |
| F | SimATTA | 23.70 | 33.82 | 26.11 | 23.55 | 23.36 | 40.16 | 43.41 | 30.22 | 41.84 | 26.42 | 40.72 | 2.88 | 41.37 | 49.21 | **52.85** | 33.31 |
| B | BATTA-RL | 33.16 | 37.75 | 28.21 | 34.97 | 26.27 | **48.57** | **49.42** | 43.11 | **47.16** | 37.84 | **51.41** | 10.01 | **47.21** | **54.03** | 52.72 | **40.12** |

(c) Tiny-ImageNet-C.

# D EXPERIMENT DETAILS

We conducted all experiments with three random seeds [0, 1, 2] and reported the mean and standard deviation values. The experiments were mainly conducted on NVIDIA RTX 3090 and TITAN GPUs, where BATTA-RL consumed 5 minutes on PACS.

## D.1 SETTINGS

**Dataset.** We utilized the corruption dataset (CIFAR10-C, CIFAR100-C, Tiny-ImageNet-C (Hendrycks & Dietterich, 2019)) and domain generalization baselines (PACS (Li et al., 2017)). CIFAR10-C/CIFAR100-C/Tiny-ImageNet-C is a 10/100/200-class dataset of a total of 150,000 images in 15 types of image corruptions, including Gaussian, Snow, Frost, Fog, Brightness, Contrast, Elastic Transformation, Pixelate, and JPEG Compression. PACS is a 7-class dataset with 9,991 images in four domains of art painting, cartoon, photo, and sketch.

Table 10: Accuracy (%) and standard deviation comparisons with **original TTA and full-label active TTA baselines** in PACS. The domain-wise data stream is a continual TTA setting (Wang et al., 2022), and the mixed data stream shuffled all domains randomly, where we report the cumulative accuracy at each of the four adaptation points. F: Full-label feedback active TTA, B: Binary-feedback active TTA. Results outperforming all other baselines are highlighted in **bold** fonts. Averaged over three random seeds.

| Label | Method | Domain-wise data stream | | | | Mixed data stream | | | |
|---|---|---|---|---|---|---|---|---|---|
| | | Art | Cartoo- | Sketch | Avg | 25% | 50% | 75% | 100%(Avg) |
| - | SrcValid | 59.38 ±0.00 | 27.94 ±0.21 | 42.96 ±0.01 | 43.43 ±0.07 | 42.74 ±1.13 | 42.80 ±0.22 | 42.64 ±0.30 | 42.77 ±0.01 |
| - | BN Stats | 67.87 ±0.18 | 63.48 ±0.88 | 54.07 ±0.36 | 61.81 ±0.18 | 59.09 ±0.29 | 58.28 ±0.08 | 58.05 ±0.22 | 57.82 ±0.20 |
| - | TENT | 71.61 ±0.70 | 67.00 ±0.51 | 44.14 ±0.85 | 60.92 ±0.29 | 60.34 ±0.51 | 56.75 ±0.62 | 53.22 ±0.57 | 49.64 ±0.50 |
| - | EATA | 68.44 ±0.31 | 64.90 ±0.69 | 58.58 ±0.18 | 63.97 ±0.23 | 59.60 ±0.15 | 58.98 ±0.54 | 59.10 ±0.38 | 59.24 ±0.08 |
| - | SAR | 67.90 ±0.20 | 63.60 ±0.83 | 55.23 ±0.44 | 62.25 ±0.11 | 59.13 ±0.21 | 58.49 ±0.15 | 58.32 ±0.05 | 58.25 ±0.07 |
| - | CoTTA | 67.87 ±0.18 | 63.48 ±0.88 | 54.06 ±0.35 | 61.81 ±0.19 | 59.10 ±0.32 | 58.29 ±0.09 | 58.06 ±0.23 | 57.83 ±0.22 |
| - | RoTTA | 64.39 ±0.59 | 38.27 ±0.61 | 40.80 ±1.64 | 47.82 ±0.20 | 52.64 ±0.25 | 49.01 ±0.85 | 46.87 ±0.55 | 45.75 ±0.49 |
| - | SoTTA | 69.86 ±0.78 | 32.02 ±1.52 | 23.66 ±1.77 | 41.84 ±0.34 | 51.96 ±5.47 | 49.84 ±6.14 | 48.09 ±6.64 | 47.06 ±6.03 |
| F | SimATTA | **77.13 ±0.76** | 71.46 ±2.47 | **78.80 ±0.53** | **75.80 ±0.74** | **68.27 ±1.24** | **72.67 ±0.45** | **75.41 ±0.30** | **77.47 ±0.44** |
| B | BATTA-RL | 73.86 ±3.76 | **76.81 ±2.45** | 76.03 ±1.61 | 75.57 ±0.93 | 59.65 ±0.70 | 64.70 ±0.78 | 69.23 ±0.17 | 72.18 ±0.38 |

Table 11: Accuracy (%) comparisons with **enhanced TTA on fine-tuned model** and binary-feedback active TTA baselines on source model, in CIFAR10-C (severity level 5). Notation * indicates the modified algorithm to utilize binary-feedback samples. E: Enhanced TTA, B: Binary-feedback active TTA. Results outperforming all other baselines are highlighted in **bold** fonts. Averaged over three random seeds.

| Label | Method | Noise | | | Blur | | | | Weather | | | | Digital | | | | |
|---|---|---|---|---|---|---|---|---|---|---|---|---|---|---|---|---|---|---|
| | | Gau. | Shot | Imp. | Def. | Gla. | Mot. | Zoom | Snow | Fro. | Fog | Brit. | Cont. | Elas. | Pix. | JPEG | Avg. |
| E | SrcValid | 76.17 | 77.48 | 67.54 | 82.24 | 71.89 | 79.90 | 83.44 | 82.67 | 84.36 | 81.18 | 88.74 | 75.12 | 77.53 | 80.66 | 80.24 | 79.28 |
| E | BN-Stats | 77.90 | 79.66 | 86.52 | 73.53 | 85.26 | 86.77 | 84.66 | 85.27 | 84.07 | 90.10 | 86.70 | 79.39 | 84.76 | 78.98 | 82.36 |
| E | TENT | 77.52 | 76.94 | 63.79 | 68.35 | 52.67 | 56.00 | 55.58 | 52.93 | 49.02 | 45.02 | 43.94 | 33.46 | 32.12 | 31.39 | 29.27 | 51.20 |
| E | EATA | 77.18 | 75.32 | 64.66 | 70.73 | 58.46 | 64.62 | 70.22 | 68.00 | 68.34 | 67.35 | 75.81 | 69.52 | 62.93 | 69.02 | 64.28 | 68.43 |
| E | SAR | 77.90 | 79.66 | 86.52 | 73.53 | 85.26 | 86.77 | 84.66 | 85.27 | 84.07 | 90.10 | 86.70 | 79.39 | 84.76 | 78.98 | 82.36 |
| E | CoTTA | 77.90 | 79.66 | 71.77 | 86.52 | 73.53 | 85.26 | 86.77 | 84.66 | 85.27 | 84.06 | 90.09 | 86.71 | 79.39 | 84.76 | 78.98 | 82.36 |
| E | RoTTA | 78.93 | 81.00 | 74.28 | 86.56 | 75.45 | 86.18 | 88.63 | 86.85 | 87.71 | 86.73 | 91.36 | 88.06 | 82.41 | 87.19 | 82.42 | 84.25 |
| E | SoTTA | **79.19** | 81.45 | 74.23 | 82.67 | 70.73 | 81.99 | 85.41 | 82.78 | 83.69 | 85.02 | 89.40 | 84.41 | 78.41 | 83.44 | 78.94 | 81.45 |
| B | SimATTA* | 76.21 | 80.88 | 74.07 | 82.17 | 73.65 | 81.70 | 85.93 | 83.17 | 86.21 | 83.08 | 90.55 | 75.75 | 81.09 | 84.65 | 84.22 | 81.56 |
| B | BATTA-RL | 76.78 | **84.24** | **78.75** | **87.51** | **77.39** | **88.38** | **91.36** | **89.42** | **90.72** | **90.30** | **94.65** | **92.62** | **86.15** | **92.42** | **87.24** | **87.20** |

**Source domain pre-training.** We closely followed the settings and utilized the pre-trained weights provided by SoTTA (Gong et al., 2023b) and SimATTA (Gui et al., 2024). As the backbone model, we employ the ResNet18 (He et al., 2016) from TorchVision (maintainers & contributors, 2016). For CIFAR10-C/CIFAR100-C/Tiny-ImageNet-C, we trained the model with the source data with a learning rate of 0.1/0.1/0.001 and a momentum of 0.9, with cosine annealing learning rate scheduling for 200 epochs. For PACS, we fine-tuned the pre-trained weights from ImageNet on the selected source domains for 3,000 iterations using the Adam optimizer with a learning rate of 0.0001.

**Scenario.** For the number of binary-feedback samples, we used $k = 3$ samples from a 64-sample test batch, accounting for less than 5% of the total data size. For the binary version of TTA baselines, we added cross-entropy loss (for correct samples) combined with complementary loss (for incorrect samples, Kim et al. (2019)), maintaining an equal budget size to our method. To implement, we replace the original TTA loss $l_{TTA}$ with $l_{TTA} + l_{CE} + l_{CCE}$, where $l_{CE}$ is a cross-entropy loss on correct samples and $l_{CCE} = -\sum_{k=1}^{num\_class} y_k \log(1 - f_\theta(k|x))$ is the complementary cross-entropy loss (Kim et al., 2019) on incorrect samples. For enhanced TTA, we used the same binary version loss with an SGD optimizer with a learning rate of 0.001 and a batch size of 64. The number of fine-tuning epochs was set to 150 for PACS, 150 for CIFAR-10, 150 for CIFAR-100, and 25 for Tiny-ImageNet-C. Note that the hyperparameters were selected to maximize accuracy on the test data stream, which is unrealistic since test data stream accuracy is not accessible during the fine-tuning process.

**Comparison with active TTA.** To compare BATTA-RL with full-label feedback methods, we propose two scenarios: (1) an equal labeling cost and (2) an equal number of active samples. To

compare with an equal labeling cost, we formulate the labeling cost with Shannon information gain (MacKay, 2003) as $\log(p^{-1})$ where $p$ is the probability of selecting a label. We assume the probability of each feedback strategy as $p = 2^{-1}$ (correct/incorrect) and $p = \texttt{num\_class}^{-1}$ (select in the entire class set). The final labeling cost for binary feedback is 1 for binary feedback and $\log(\texttt{num\_class})$ for full-label feedback. Therefore, we utilize $\log(\texttt{num\_class})$ times more feedback samples for BATTA setting compared to active TTA.

### D.2 TTA BASELINES

**TENT.** For TENT (Wang et al., 2021), we utilize an Adam optimizer (Kingma & Ba, 2015) with a learning rate $LR = 0.001$, aligning with the guidelines outlined in the original paper and active TTA paper (Gui et al., 2024). The implementation followed the official code.[1]

**EATA.** For EATA (Niu et al., 2022), we followed the original configuration of $LR = 0.001$, entropy constant $E_0 = 0.4 \times \ln C$, where $C$ represents the number of classes. Additionally, we set the cosine sample similarity threshold $\epsilon = 0.5$, trade-off parameter $\beta = 2,000$, and moving average factor $\alpha = 0.1$. The Fisher importance calculation involved 2,000 samples, as recommended. The implementation followed the official code.[2]

**SAR.** For SAR (Niu et al., 2023), we set a learning rate of $LR = 0.00025$, sharpness threshold $\rho = 0.5$, and entropy threshold $E_0 = 0.4 \times \ln C$, following the recommendations from the original paper. The top layer (layer 4 for ResNet18) was frozen, consistent with the original paper. The implementation followed the official code.[3]

**CoTTA.** For CoTTA (Wang et al., 2022), we set the restoration factor $p = 0.01$, and exponential moving average (EMA) factor $\alpha = 0.999$. For augmentation confidence threshold $p_{th}$, we followed the previous implementation (Gui et al., 2024) as $p_{th} = 0.1$. The implementation followed the official code.[4]

**RoTTA.** For RoTTA (Yuan et al., 2023), we utilized the Adam optimizer (Kingma & Ba, 2015) with a learning rate of $LR = 0.001$ and $\beta = 0.9$. We followed the original hyperparameters, including BN-statistic exponential moving average updating rate $\alpha = 0.05$, Teacher model's exponential moving average updating rate $\nu = 0.001$, timeliness parameter $\lambda_t = 1.0$, and uncertainty parameter $\lambda_u = 1.0$. The implementation followed the original code.[5]

**SoTTA.** For SoTTA (Gong et al., 2023b), we utilized the Adam optimizer (Kingma & Ba, 2015), with a BN momentum of $m = 0.2$ and a learning rate of $LR = 0.001$. The memory size was set to 64, with the confidence threshold $C_0 = 0.99$. The entropy-sharpness L2-norm constraint $\rho$ was set to 0.5, aligning with the suggestion (Foret et al., 2021). The top layer was frozen following the original paper. The implementation followed the original code.[6]

**SimATTA.** We follow the original implementation of SimATTA (Gui et al., 2024). Since SimATTA queries active samples at a dynamic rate, we set the centroid increase number to $k = 3$ and limit the budget per batch to 3, ensuring an equal active sample budget compared to BATTA-RL. For the adaptation objective, we add the complementary loss (incorrect samples, Kim et al. (2019)) to the original cross-entropy loss for correct samples. For CIFAR-10 and CIFAR-100, we performed a grid search to find the optimal hyperparameters. We found the optimal hyperparameters to be $LR = 0.0001/0.0001$, $e_h = 0.001/0.001$, and $e_l = 0.0001/0.00001$ for the CIFAR-10 and CIFAR-100 datasets, respectively. The implementation is based on the original code.[7]

---

[1] https://github.com/DequanWang/tent
[2] https://github.com/mr-eggplant/EATA
[3] https://github.com/mr-eggplant/SAR
[4] https://github.com/qinenergy/cotta
[5] https://github.com/BIT-DA/RoTTA
[6] https://github.com/taeckyung/sotta
[7] https://github.com/divelab/ATTA

**BATTA-RL (Ours).** We utilize an SGD optimizer with a learning rate/epoch of 0.001/3 (CIFAR10-C, PACS), 0.0001/3 (CIFAR100-C), and 0.00005/5 (Tiny-ImageNet-C) on the entire model. We applied stochastic restoration (Wang et al., 2022) in Tiny-ImageNet-C to prevent overfitting. We update batch norm statistics with the unlabeled test batch before active labeling and freeze the statistics during adaptation, following Gui et al. (2024). We apply the dropout layer after residual blocks, following the previous work on TTA accuracy estimation (Lee et al., 2024b), with a dropout rate of 0.3, except for 0.1 on Tiny-ImageNet-C. Additionally, we introduce a memory mechanism to enhance adaptation stability. We maintain a record of recent binary feedback samples (each correct and incorrect) with a memory size equal to the batch size. Then, we calculate the mean gradient from each correct and incorrect sample memory and sum them up. This ensures the balancing between correct and incorrect samples in the early stage, where the number of each sample is imbalanced. After filling the memory up, the summation is equivalent to $\alpha \mathbb{E}_{x \in \mathcal{S}_{\text{BFA}}}[R_{\text{BFA}}(x, y) \nabla_\theta \log \pi_\theta(y|x)]$ with $\alpha = 2$.

## E    LICENSE OF ASSETS

**Datasets.** CIFAR10-C/CIFAR100-C (Creative Commons Attribution 4.0 International), and Tiny-ImageNet-C dataset (Apache-2.0). The license of PACS dataset is not specified.

**Codes.** Torchvision for ResNet18 (Apache 2.0), the official repository of TENT (MIT License), the official repository of EATA (MIT License), the official repository of SAR (BSD 3-Clause License), the official repository of CoTTA (MIT License), the official repository of RoTTA (MIT License), the official repository of SoTTA (MIT License), and the official repository of SimATTA (GPL-3.0 License).

