# OpenReview forum: "Binary-Feedback Active Test-Time Adaptation"
_ICLR.cc/2025/Conference — Submitted to ICLR 2025_

### Official Review · Reviewer_ZvDw · 2024-10-30

**Soundness:** 3
**Presentation:** 3
**Contribution:** 3
**Rating:** 8
**Confidence:** 4

**Summary:**

Deep learning models often face performance issues when there are domain shifts between training and test data. since Test-Time Adaptation (TTA) methods have limitations, and active TTA approaches with full-class labels are impractical due to high costs, this paper proposes a Binary-feedback Active Test-Time Adaptation (BATTA) setting and a method named BATTA-RL to address the performance degradation of deep learning models due to domain shifts when testing. Specifically, in BATTA, an oracle provides binary feedback (correct/incorrect) on model predictions. This feedback is integrated in real-time to guide continuous model adaptation. A dual-path optimization framework is proposed to leverage binary feedback and unlabeled samples, which is balanced by reinforcement learning. Experimental results demonstrates the effectiveness of the proposed methods.

**Strengths:**

1. This paper proposes a novel and practical active learning test-time adaptation paradigm. It presents an innovative approach that combines binary feedback with unlabeled samples, effectively addressing the issue of domain shifts in deep learning models when testing. In this setting, an annotator provides binary feedback (indicating whether a model prediction is correct or incorrect) instead of full-class labels. This reduces the labeling burden significantly as binary feedback requires less information compared to full - class labels. For example, the human effort and error rate in providing binary feedback are much lower than in full-class labeling as demonstrated by previous studies.

2. The proposed method, BATTA-RL, shows promising results in improving model performance with minimal labeling effort, which is a significant contribution to the field of test-time adaptation.

3. The experimental setup is comprehensive, and the comparisons with existing methods are thorough, providing strong evidence of the superiority of the proposed paradigm.

**Weaknesses:**

1. It would be beneficial for the article to further explain why the reward functions for the two paths are set differently. For example, in Binary Feedback - Guided Adaptation (BFA), the reward function value is -1 in the incorrect case, while in Agreemend - Based Self - Adaptation (ABA), the reward function value is 0 in the case of disagreement. What are the considerations behind such designs? This clarification would enhance the understanding of the proposed method and its underlying mechanisms.

2. It would be advisable for the article to further explain in the appendix, using formulas if possible, how the baselines of previous TTA and ATTA are adapted to fit into the proposed setting to ensure fairness. This would provide more transparency and a deeper understanding of the experimental setup and comparisons made in the study.

**Questions:**

Please refer to Weaknesses.

---

> ### Author Response · Authors · 2024-11-20
> **Response to Reviewer ZvDw**
>
> Dear Reviewer ZvDw,
>
> We sincerely appreciate your constructive feedback on our work. We have substantially revised the paper to address all raised concerns and strengthen our contribution. We provide detailed responses to each point:
>
> > W1. Considerations behind reward function design for BFA and ABA.
>
> - We thank the reviewer for this insightful question about our reward function design. The different reward structures for BFA and ABA were carefully chosen based on the reliability of each signal. In BFA, we have explicit binary feedback from the oracle, so we can confidently assign a negative reward (-1) to incorrect predictions to actively discourage them. However, in ABA, prediction disagreement doesn't necessarily indicate incorrect predictions - as demonstrated in Figure 4(b), disagreement samples show mixed accuracy rather than consistent incorrectness. Therefore, rather than actively penalizing these potentially noisy signals with negative rewards, we simply excluded them from the adaptation process by setting their reward to 0. This design choice allows our method to leverage confident predictions while gracefully handling uncertain cases without introducing potentially harmful adaptation signals. We added this clarification to the paper to better explain the rationale behind our reward function design.
>
> > W2. How the baselines of previous TTA and ATTA are adapted to fit into the proposed setting to ensure fairness.
>
> - We appreciate this important point about experimental transparency. We have substantially expanded the implementation details in Appendix D.2 to explain how we adapted existing TTA and ATTA methods to incorporate binary feedback. For TTA baselines, we modified their objectives to:
> $L = L_{TTA} + L_{CE} + L_{CCE}$,
> where L_TTA is the original TTA loss, L_CE is the cross-entropy loss on correct samples, and L_CCE is the complementary cross-entropy loss on incorrect samples following Kim et al. [1]:
> $L_{CCE} = -∑^{num\_class}_{k=1}  y_k \log (1 -  f_θ ( k | x ) ) $.
> For the active TTA baseline (SimATTA), we adapted its clustering-based sample selection while modifying its supervision signal to use both correct and incorrect binary feedback using the same loss formulation above.
> The complete implementation details, including hyperparameter settings, are now documented in Appendix D.2.
>
> [1] Youngdong Kim, Junho Yim, Juseung Yun, and Junmo Kim. Nlnl: Negative learning for noisy labels.
> In Proceedings of the IEEE/CVF international conference on computer vision, 2019.

---

> > ### Comment · Reviewer_ZvDw · 2024-11-25
> >
> > Thank you for your response. My concern has been addressed.

---

> > > ### Author Response · Authors · 2024-11-25
> > >
> > > We sincerely appreciate your time and effort in reviewing our manuscript. Your thoughtful feedback and constructive suggestions have helped strengthen our work.
> > >
> > > We are grateful for your recognition of our contributions to the field. We would be happy to continue the discussions if you have any further questions.
> > >
> > > Best regards,
> > >
> > > Authors.

---

### Official Review · Reviewer_afeQ · 2024-11-01

**Soundness:** 2
**Presentation:** 3
**Contribution:** 3
**Rating:** 6
**Confidence:** 4

**Summary:**

The paper introduces Binary-feedback Active Test-Time Adaptation (BATTA), a novel TTA setting for adapting deep learning models to domain shifts at test time using binary feedback from human annotators. The authors address limitations in prior active TTA methods, which suffer from high annotation costs, especially in multi-class settings. To mitigate this, they propose BATTA-RL, a reinforcement learning-based approach with a dual-path optimization strategy. BATTA-RL combines Binary Feedback-guided Adaptation (BFA) for uncertain samples and Agreement-Based Self-Adaptation (ABA) for confident samples, enhancing model performance on challenging test distributions. Experiments across multiple datasets demonstrate that BATTA-RL outperforms existing TTA methods.

**Strengths:**

1. **Reduced Labeling Costs**: BATTA minimizes labeling demands by using binary feedback from human annotators instead of requiring full-class labels. This significantly reduces the annotation burden, making it more feasible for real-world scenarios than current ATTA methods.

2. **Dual-Path Optimization with Reinforcement Learning**: BATTA-RL combines Binary Feedback-guided Adaptation (BFA) for uncertain samples and Agreement-Based Self-Adaptation (ABA) for confident samples. Introducing reinforcement learning by binary feedback optimization is interesting and novel in TTA.

3. **Strong Experimental Results**: BATTA-RL consistently outperforms competing TTA methods and even surpasses the ATTA method with full-class labels.

**Weaknesses:**

1. **Experimental Setting**: Appendix D.1 mentions multiple epochs of adaptation for all datasets. However, in TTA settings—where real-time, stream-based tasks are required—multiple epochs of adaptation are impractical. This approach hinders real-time inference capabilities.

2. **Scalability of BATTA**: Without extensive experimentation on large-scale datasets, the scalability of BATTA remains uncertain. Testing on a dataset like ImageNet-C with 1000 classes would be insightful. In scenarios with high model error rates, where most feedback might be "incorrect," there is a risk of BATTA collapsing. Additionally, as shown in Table 1(c), BATTA only surpasses SAR by less than 0.3%, which is a marginal improvement.

3. **Frequency of Human Intervention**: The requirement of annotating 3 samples per batch of 64 suggests a substantial annotation budget, even with binary feedback. The need for human intervention in every test batch implies that annotators must be continuously available while the system operates. Reducing the annotation budget and frequency could make the system more practical.

4. **Computational Complexity**: Monte Carlo dropout likely introduces additional computational demands. The manuscript lacks a thorough comparison of computational complexity against competing methods. The statement that “experiments were mainly conducted on NVIDIA RTX 3090 and TITAN GPUs, with BATTA-RL consuming 5 minutes on PACS” is vague. Specific hardware details and wall-clock time comparisons with other methods, per test sample, are necessary for clarity.

**Questions:**

Please refer to **Weakness**.

---

> ### Author Response · Authors · 2024-11-20
> **Response to Reviewer afeQ**
>
> Dear Reviewer afeQ,
>
> We sincerely appreciate your constructive feedback on our work. We have substantially revised the paper to address all raised concerns and strengthen our contribution. We provide detailed responses to each point:
>
> > W1. The use of multiple epochs hinders real-time inference capabilities.
>
> - We appreciate this important concern about real-time practicality. While our main results use multiple epochs to demonstrate BATTA-RL's full capability, we conducted additional experiments examining performance under smaller epochs. Our analysis in the table below shows BATTA-RL maintains strong performance even with single-epoch adaptation:
> On CIFAR10-C, reducing from 3 epochs to 1 epoch (with proportionally adjusted learning rate ×3) achieves 87.11% accuracy compared to 87.20% with 3 epochs. Similar robust performance is observed with 2 epochs (87.15%). This demonstrates that BATTA-RL can effectively adapt in scenarios where multiple epochs are impractical.
> We have added these findings in Section B of the Appendix, with results as in the table below (corresponding to Table 6 in Appendix B) showing consistent performance across different corruption types under single-epoch adaptation. These results suggest BATTA-RL is viable for real-time applications while maintaining its advantages over existing methods.
>
> | Method | Avg. |
> |---------|------|
> | BATTA-RL (epoch = 3) | 87.20 |
> | · epoch = 1 | 87.11 |
> | · epoch = 2 | 87.15 |
>
> Table R6. Accuracy (%) comparisons with various epoch settings.
>
>
> > W2. Scalability of BATTA under large-scale datasets (e.g., ImageNet-C) and marginal improvement in Tiny-ImageNet-C.
>
> - Thank you for the suggestion. Our experiment setting is acknowledged to be comprehensive (Reviewer ZvDw), and we understand that adding further experiments would strengthen our paper.
>
> - We conducted a thorough additional experiment and discussed in the global response. Superior performance on various large-scale datasets, including ImageNet-C, demonstrates the effectiveness of our dual-path optimization framework. Please check our global response to address your concern.

---

> ### Author Response · Authors · 2024-11-20
> **Response to Reviewer afeQ (Part II)**
>
> > W3. Frequency of Human Intervention: The requirement of annotating 3 samples per batch of 64 suggests a substantial annotation budget, even with binary feedback. The need for human intervention in every test batch implies that annotators must be continuously available while the system operates. Reducing the annotation budget and frequency could make the system more practical.
>
> We thank the reviewer for raising this important practical consideration about annotation frequency. Our experiments demonstrate BATTA-RL's robustness across various annotation frequencies, maintaining strong performance with as few as 1-2 binary feedbacks per batch (85.02-86.23% accuracy) and during intermittent labeling scenarios where annotations are only available for a fraction of batches (84.56-85.83% accuracy).
>
> - First, our setting of 3 binary-feedback samples per batch of 64 (less than 5%) was chosen to match the sample budget used in previous active TTA work [1] for a fair comparison. Importantly, binary feedback is significantly less intensive than full-label annotation - requiring only a yes/no response versus selecting from all possible classes.
>
> - Second, as shown in Figure 7 in the manuscript, BATTA-RL maintains strong performance compared to the baseline (SimATTA*) in varying numbers of annotations per batch. We additionally conducted 1-2 binary feedbacks per batch, and the results demonstrated that BATTA-RL remains effective with a few samples. We updated Figure 7 in the manuscript correspondingly.
>
> | Method | 0 (no feedback) | 1 | 2 | 3 |
> |---------|----------|---------|-----------|-----------|
> | SimATTA* | 57.03 | 75.53 | 79.34 | 81.09 |
> | BATTA-RL | 82.64 | 85.02 | 86.23 | 87.20 |
>
> Table R7. Accuracy (%) comparisons with different numbers of annotations per batch.
>
>
> -  Furthermore, we conducted new experiments during the rebuttal period to evaluate scenarios where annotators are not continuously available. We set the scenario with intermittent labeling where the labels are only partially available (for ½, ⅓, and ¼ of total batches). As in the table below (corresponding to Figure 11 in Appendix B), compared to the baseline (SimATTA*), our BATTA-RL maintains robust performance even with intermittent human feedback.
>
> | Method | 0 Skips | 1 Skip | 2 Skips | 3 Skips |
> |---------|----------|---------|-----------|-----------|
> | SimATTA* | 81.09 | 77.61 | 76.13 | 73.96 |
> | BATTA-RL | 87.20 | 85.83 | 84.98 | 84.56 |
>
> Table R8. Accuracy (%) comparisons with different numbers of labeling skipped batches.
>
> - These results demonstrate our method's flexibility in balancing performance and annotation frequency based on practical constraints. We agree that exploring even more annotation-efficient strategies is an important direction for future work and would be happy to expand this discussion in the paper.
>
> [1] Shurui Gui, Xiner Li, and Shuiwang Ji. Active test-time adaptation: Theoretical analyses and an algorithm. In International Conference on Learning Representations (ICLR), 2024.
>
>
> > W4. Computational complexity of BATTA-RL.
>
> - We thank the reviewer for this important practical concern. Please check our new wall-clock time analysis in the global response.

---

> > ### Comment · Reviewer_afeQ · 2024-11-25
> >
> > Thank you to the authors for your detailed response. I appreciate the additional analysis on ImageNet-C and the consideration of sparser human intervention. These additions meaningfully strengthen the manuscript and effectively address my primary concerns.
> >
> > Overall, great rebuttal. I am pleased to raise my rating.

---

> > > ### Author Response · Authors · 2024-11-25
> > >
> > > Thank you for upgrading your score! We sincerely appreciate your time and effort in reviewing our manuscript. Your thoughtful feedback and constructive suggestions have helped strengthen our work.
> > >
> > > We are pleased that our rebuttal has addressed your concerns. We would be happy to continue the discussion if you have any further questions.
> > >
> > > Best regards,
> > >
> > > Authors.

---

### Official Review · Reviewer_Pxu6 · 2024-11-03

**Soundness:** 3
**Presentation:** 3
**Contribution:** 3
**Rating:** 6
**Confidence:** 3

**Summary:**

Motivated by the high annotation budget of the previous active TTA, where an oracle provides accurate ground-truth labels for selected samples, this paper defines a more realistic active test-time adaptation setting (binary feedback active TTA) with relatively weak assumptions. To achieve this, the authors propose an RL framework, BATTA-RL, consisting of Binary Feedback-guided Adaptation (BFA) and Agreement-based Self-Adaptation (ABA). BFA is proposed to learn from the valuable feedback information and ABA is proposed to improve self-training with MC dropout predictions. The evaluation experiments are conducted on CIFAR10-C, CIFAR100-C and Tiny-ImageNet-C. The results show the effectiveness and excellent improvement of the proposed method.

**Strengths:**

1. This paper introduces reinforcement learning to test-time adaptation and provides the unsupervised TTA method with low-cost feedback to improve the robustness of the TTA method.
2. The paper is well written and easy to follow, and the experiments on several benchmark datasets validate its effectiveness.

**Weaknesses:**

1. Using the ensemble predictions of multiple data or feature augmentation (e.g., dropout) to estimate the certainty of the samples or to obtain the robust predictions is already well known in the TTA tasks, such as MEMO[A] and CoTTA[B]. One suggestion for improvement would be for the authors to compare different uncertainty estimation strategies, e.g. those used in MEMO and CoTTA.
2. Using the ensemble predictions to obtain the uncertainty of the samples could be more time consuming. It's better for the authors to calculate the wall clock time for the proposed method and compare it with others.
3. To demonstrate the effectiveness of the proposed sample selection method used in BFA, a comparison with random selection is necessary. What is the performance of the BFA module using feedback from randomly selected test samples rather than those from top-k uncertainty samples?

[A] MEMO: Test Time Robustness via Adaptation and Augmentation
[B] Continual Test-Time Domain Adaptation

**Questions:**

See the weakness.

---

> ### Author Response · Authors · 2024-11-20
> **Response to Reviewer Pxu6**
>
> Dear Reviewer Pxu6,
>
> We sincerely appreciate your constructive feedback on our work. We have substantially revised the paper to address all raised concerns and strengthen our contribution. We provide detailed responses to each point:
>
> > W1. Compare with feature augmentation-based uncertainty estimation.
>
> - We appreciate this insightful suggestion about comparing uncertainty estimation strategies. Following your recommendation, we conducted additional experiments comparing our MC dropout-based uncertainty estimation with augmentation-based approaches used in MEMO and CoTTA. Our findings reveal that augmentation-based uncertainty estimation leads to severe performance degradation (17% accuracy) in the BATTA setting, compared to our MC dropout approach (87.20% on CIFAR10-C).
> This substantial performance gap reveals a critical limitation: augmentation-based uncertainty estimates tend to overfit in the early adaptation stage, making them unreliable for active sample selection in our binary feedback setting. In contrast, MC dropout provides more stable uncertainty estimates by directly capturing the model's epistemic uncertainty, leading to more reliable sample selection for binary feedback queries.
> We have added these comparative results and detailed analysis in Appendix B, which demonstrate why MC dropout is better suited for uncertainty estimation in the binary feedback setting.
>
> | Method | Avg. |
> |---------|-------|
> | BATTA-RL (original) | 87.20 |
> | Augmentation-based | 19.07 |
>
> Table R5. Accuracy (%) comparisons of BATTA-RL original version and augmentation-based uncertainty estimation.
>
>
>
> > W2. Wall-clock time comparison of the method.
>
> - We thank the reviewer for this important practical concern. Please check our new wall-clock time analysis in the global response.
>
>
> > W3. Impact of sample selection for BFA in BATTA-RL.
>
> - We agree with this important suggestion about validating our sample selection strategy. We have findings in Figure 9 in Appendix B of our paper - we comprehensively evaluated various sample selection strategies, including random selection, maximum entropy, minimum confidence, and minimum energy, against our MC-dropout uncertainty-based selection. The results demonstrate that our approach significantly outperforms random selection and other baseline strategies in CIFAR10-C. This empirically validates that selecting samples based on MC-dropout uncertainty is more effective than random selection for guiding the adaptation process. We believe these results provide strong evidence for the effectiveness of our sample selection method in the BFA module.

---

> > ### Comment · Reviewer_Pxu6 · 2024-12-01
> >
> > Thank you for the detailed response! All my concerns have been addressed and I will keep my current rating.

---

> > > ### Author Response · Authors · 2024-12-02
> > >
> > > We sincerely appreciate your time and effort in reviewing our manuscript. Your thoughtful feedback and constructive suggestions have helped strengthen our work.
> > >
> > > We are grateful that all your concerns have been addressed. We would be happy to continue the discussions if you have any further questions.
> > >
> > > Best regards,
> > >
> > > Authors.

---

### Official Review · Reviewer_8qrn · 2024-11-04

**Soundness:** 3
**Presentation:** 3
**Contribution:** 3
**Rating:** 5
**Confidence:** 3

**Summary:**

The paper proposes using binary feedback for active test-time adaptation (TTA) through reinforcement learning. It begins by employing MC-dropout to perform multiple forward passes. Based on the softmax output from MC-dropout, uncertain samples (those with low confidence) are selected and sent to an annotator for binary feedback. This feedback is then utilized in the learning process through the REINFORCE loss. For more certain samples, an Agreement-Based self-Adaptation (ABA) module is implemented, encouraging the model to maintain consistent predictions via a reward signal.

**Strengths:**

1. The concept of using binary feedback to guide the model is intriguing.
2. Utilizing MC-dropout to estimate sample confidence is also a promising approach.
3. Experimental results demonstrate significant improvements in model performance.

**Weaknesses:**

1. The signal acquisition process may be impractical; relying on an annotator could be costly or unrealistic in certain settings. However, advancements in foundation models may help address this issue soon.
2. The computational cost of MC-dropout is notable. Since multiple inference steps are required to gather this information, its application to TTA may raise efficiency concerns. Reviewers expect the authors to include experiments that evaluate the running time of the proposed method.
3. The fairness of the experiments is questionable. Given that this approach utilizes multiple MC-dropout steps, it effectively functions as a form of ensemble prediction, which naturally enhances performance. In contrast, other methods typically rely on a single forward pass.

**Questions:**

Please refer to the Weakness

---

> ### Author Response · Authors · 2024-11-20
> **Response to Reviewer 8qrn**
>
> Dear Reviewer 8qrn,
>
> We sincerely appreciate your constructive feedback on our work. We have substantially revised the paper to address all raised concerns and strengthen our contribution. We provide detailed responses to each point:
>
> > W1. The signal acquisition process may be impractical; relying on an annotator could be costly or unrealistic in certain settings. However, advancements in foundation models may help address this issue soon.
>
> - We truly agree that end-user annotation is costly and limits the applicability of full-label active TTA. This concern about labeling burden motivated our development of binary-feedback active TTA (BATTA) as a more practical alternative to full-label annotation. The novelty and usefulness of the BATTA setting are recognized by Reviewer afeQ and ZvDw.
>
> - We strongly agree with the reviewer's insight about leveraging foundation models as oracles - this represents an exciting future direction that could further reduce dependency on human annotators while maintaining the benefits of our binary feedback framework. We plan to explore this direction in future work and would welcome the opportunity to discuss it further in the paper.
>
> > W2. The computational cost of MC-dropout.
>
> - We appreciate this important concern about MC dropout's computational overhead. Please check our new wall-clock time analysis in the global response.
>
>
> > W3. The fairness of using multiple forward passes.
>
> - We thank the reviewer for raising this important point about experimental fairness. We respectfully disagree that our use of MC-dropout creates an unfair advantage. Most state-of-the-art TTA methods also employ various forms of ensemble or multiple forward passes: SAR/SoTTA require double forward and backward passes for sharpness-aware minimization, and CoTTA/RoTTA use multiple augmentation-based forward passes with a teacher-student framework with multiple predictions. Considering the strongest TTA baselines are using multiple forward passes, we believe our experimental comparisons are fair and meaningful.

---

> > ### Author Response · Authors · 2024-12-02
> > **Follow-up on Rebuttal Response**
> >
> > Dear Reviewer 8qrn,
> >
> > Based on your valuable feedback, we have carefully addressed your concerns. As we approach the end of the discussion period, we would greatly appreciate if you could review our response and consider our revised manuscript. Your constructive comments have helped us significantly improve our work, and we believe we have thoroughly addressed your concerns.
> >
> > Thank you for your time and detailed review. We welcome any additional questions or requests for clarification.
> >
> > Best regards,
> >
> > Authors

---

### Official Review · Reviewer_uYWh · 2024-11-10

**Soundness:** 2
**Presentation:** 2
**Contribution:** 2
**Rating:** 5
**Confidence:** 4

**Summary:**

BATTA is a novel approach for active test-time adaptation that uses binary feedback from a human to adapt pre-trained models to new domains, allowing reasonable labeling costs compared to methods requiring full-class labels. The BATTA-RL method integrates binary feedback on uncertain samples with self-adaptation on confident samples within a reinforcement learning framework. Extensive experiments demonstrate that BATTA-RL surpasses state-of-the-art test-time adaptation methods.

**Strengths:**

1. The approach of using reinforcement learning, instead of the commonly used (direct) entropy minimization objective or cross-entropy loss for TTA is intriguing and adds a new perspective.
2. The approach to not solely rely on human feedback but to incorporate self-adaptation on confident samples is realistic and efficient, making it a practical strategy for TTA.

**Weaknesses:**

1. The experiments conducted in this paper seem quite limited. It is common in current TTA to conduct evaluations on benchmarks such as ImageNet-C (additionally, ImageNet-R, and VisDA-2021). Also, to demonstrate that the use of reinforcement learning algorithms for active test-time adaptation is generally powerful, it would be important to include experiments in more complex scenarios, such as those proposed in SAR [1]: i) dynamic shifts in the ground-truth test label distribution leading to imbalanced distributions at each corruption, ii) single test sample scenarios, and iii) combinations of multiple distribution shifts in more challenging and biased situations.
2. Since the current experiments are conducted on relatively easier datasets like CIFAR-10 and Tiny ImageNet, it is likely that there are a relatively higher number of confident samples. However, in scenarios with fewer confident samples, such as those in ImageNet-C or ImageNet-R (severity level 5), wouldn't the burden of human feedback increase? In such cases, it appears that the performance may work similarly to SimATTA (full labeling). If so, what would be the key distinguishing factor of the proposed method?
3. Furthermore, as noted in DeYO [2], there can be cases where a model makes confident predictions due to spurious correlations. Would BATTA-RL still demonstrate outstanding performance when evaluated on benchmarks with significant spurious correlations? Including related observations or at least discussions on this point would be valuable.



[1] Niu, S., Wu, J., Zhang, Y., Wen, Z., Chen, Y., Zhao, P., & Tan, M. Towards Stable Test-time Adaptation in Dynamic Wild World. In The Eleventh International Conference on Learning Representations.


[2] Lee, J., Jung, D., Lee, S., Park, J., Shin, J., Hwang, U., & Yoon, S. Entropy is not Enough for Test-Time Adaptation: From the Perspective of Disentangled Factors. In The Twelfth International Conference on Learning Representations.

**Questions:**

Please address the concerns mentioned above in weaknesses.

---

> ### Author Response · Authors · 2024-11-20
> **Response to Reviewer uYWh**
>
> Dear Reviewer uYWh,
>
> We sincerely appreciate your constructive feedback on our work. We have substantially revised the paper to address all raised concerns and strengthen our contribution. We provide detailed responses to each point:
>
> > W1. Experiments on additional datasets (ImageNet-C, ImageNet-R, and VisDA-2021) and scenarios (test label distribution shift, single test sample, and combination of multiple distribution shifts).
>
> - Thank you for the suggestion. Although we still believe that our experiment setting is comprehensive (as acknowledged by Reviewer ZvDw), we understand that adding further experiments would strengthen our paper. We provide additional experimental results in the global response. For the combination of multiple distribution shifts (mixed data streams), we already included the experiments in Table 2 of the manuscript, demonstrating our BATTA-RL’s superior performance where distributions are mixed.
>
>
> > W2. Burden of human feedback on large-scale datasets.
>
> - While large-scale datasets may indeed have fewer confident samples, our additional experiments on ImageNet-C/R during the rebuttal period demonstrate that BATTA-RL maintains its superior performance even on these more challenging datasets. This scalability stems from BATTA-RL's unique dual-path optimization that leverages both binary feedback and unlabeled samples, unlike SimATTA, which relies solely on labeled samples.
>
> - Also, binary feedback remains significantly more efficient than full labeling. This efficiency becomes particularly evident in our experimental results - as shown in Figure 5 when controlling for equal labeling cost, BATTA-RL substantially outperforms full-labeled SimATTA across datasets, with the performance gap actually widening for datasets with more classes (9% improvement on CIFAR-10, 30% on CIFAR-100, and 32% on Tiny-ImageNet).
>
> > W3. Benchmarks with spurious correlations.
>
> - To investigate the impact of spurious correlations, we evaluated BATTA-RL on ColoredMNIST, an important benchmark in DeYO [1]. Our new experiments show BATTA-RL achieves 96.75% accuracy on ColoredMNIST, significantly outperforming standard TTA methods (45.59-82.70%) and active TTA baseline SimATTA (93.69%). This strong performance on ColoredMNIST suggests that BATTA-RL's dual-path optimization effectively mitigates the impact of spurious correlations. Binary feedback helps identify when the model relies on spurious features and produces wrong predictions, while agreement-based self-adaptation prevents overconfident predictions on misleading patterns. We have added these results in Table 3 in Section 4 of the manuscript.
>
> | Dataset | SrcValid | BN-Stats | TENT* | EATA* | SAR* | CoTTA* | RoTTA* | SoTTA* | SimATTA* | BATTA-RL |
> |---------|-----------|-----------|---------|---------|--------|----------|----------|----------|------------|------------|
> | ColoredMNIST | 50.49 | 45.59 | 44.92 | 45.59 | 45.74 | 45.60 | 48.90 | 59.45 | 93.66 | 96.75 |
>
> Table R4. Accuracy (%) comparisons on spurious correlations.
>
> [1] Lee, Jonghyun, et al. "Entropy is not enough for test-time adaptation: From the perspective of disentangled factors." arXiv preprint arXiv:2403.07366 (2024).

---

> > ### Author Response · Authors · 2024-12-02
> > **Follow-up on Rebuttal Response**
> >
> > Dear Reviewer uYWh,
> >
> > Based on your valuable feedback, we have expanded our experiments to include five datasets and two scenarios. As we approach the end of the discussion period, we would greatly appreciate if you could review our response and consider our revised manuscript. Your constructive comments have helped us significantly improve our work, and we believe we have thoroughly addressed your concerns.
> >
> > Thank you for your time and detailed review. We welcome any additional questions or requests for clarification.
> >
> > Best regards,
> >
> > Authors

---

### Author Response · Authors · 2024-11-20
**Global response**

We sincerely thank all reviewers for their thorough evaluation and constructive feedback. We are encouraged that the novelty and significance of our work have been recognized, and we appreciate the opportunity to strengthen our contribution through your valuable suggestions.

In our global response, we first highlight the paper's major contributions, then address the common concerns raised across reviews, and finally detail our revision changes for clarity. We have organized our response to help reviewers easily track how their feedback has been incorporated into the improved manuscript.

###  Major contributions

- Introducing a novel and practical active learning test-time adaptation paradigm (Reviewer uYWh, 8qrn, ZvDw, afeQ, ZvDw).
- Proposing a novel dual-path optimization algorithm with reinforcement learning (Reviewer uYWh, Pxu6, afeQ).
- Significant improvements over TTA/ActiveTTA methods with various datasets/scenarios (Reviewer 8qrn, Pxu6, afeQ, ZvDw).


### Common concern

- Additional experiments (Reviewer uYWH and afeQ): We conducted additional experiments on large-scale datasets and adaptation scenarios and summarized the results in the table below (also appears in  Table 3 in Section 4). In all datasets and scenarios under the BATTA setting, our BATTA-RL consistently outperformed the TTA and active TTA baselines. BATTA-RL's superior performance on large-scale datasets such as ImageNet-C demonstrates its effectiveness in large-scale test-time adaptation. The key insight is that BATTA-RL formulates both binary feedback and unlabeled sample adaptation as a single reinforcement learning objective, where the reward signals seamlessly guide the model's adaptation. The use of MC-dropout provides a robust uncertainty estimate while optimizing MC-dropout, which prevents the TTA model from overfitting and leads to a stable adaptation in large-scale datasets. Also, agreement-based adaptation (ABA) provides a robust adaptation with confident samples without requiring a fixed threshold.


| Dataset | SrcValid | BN-Stats | TENT* | EATA* | SAR* | CoTTA* | RoTTA* | SoTTA* | SimATTA* | BATTA-RL |
|---------|-----------|-----------|---------|---------|--------|----------|----------|----------|------------|------------|
| ImageNet-C | 14.43 | 26.88 | 0.93 | 30.87 | 35.15 | 22.55 | 26.80 | 36.02 | 19.50 | **36.59** |
| ImageNet-R | 33.05 | 35.08 | 29.10 | 37.14 | 36.64 | 35.02 | 34.35 | 31.00 | 35.63 | **38.59** |
| ColoredMNIST | 50.49 | 45.59 | 44.92 | 45.59 | 45.74 | 45.60 | 48.90 | 59.45 | 93.66 | **96.75** |
| VisDA-2021 | 27.36 | 26.46 | 20.38 | 27.82 | 27.41 | 26.46 | 27.23 | 27.71 | 22.80 | **29.30** |
| DomainNet | 54.82 | 54.41 | 18.80 | 59.49 | 57.78 | 54.40 | 56.41 | 54.82 | 58.41 | **60.85** |

Table R1. Accuracy (%) comparisons on additional datasets. Note: CoTTA results will be updated by this Sunday. --> Updated.


| Setting | SrcValid | BN-Stats | TENT* | EATA* | SAR* | CoTTA* | RoTTA* | SoTTA* | SimATTA* | BATTA-RL |
|---------|-----------|-----------|---------|---------|--------|----------|----------|----------|------------|------------|
| Imbalanced (non-iid) | 57.70 | 26.58 | 23.79 | 17.54 | 26.63 | 26.58 | 50.66 | 76.34 | 75.62 | **86.91** |
| Batch size 1 | 57.70 | 27.82 | 10.04 | 27.82 | 28.12 | 27.82 | 10.14 | 45.92 | - | **70.17** |

Table R2. Accuracy (%) comparisons on additional scenarios.



- Computational efficiency (Reviewer 8qrn, Pxu6, and afeQ): We conducted a comprehensive runtime analysis by measuring the average wall-clock time per batch across different methods on the Tiny-ImageNet-C dataset.
Our results in the table below (also appears in Table 4 in Section 4) show that BATTA-RL requires 4.19 ±0.06 seconds per batch, positioning it between simpler TTA methods (0.33-1.68s) and more complex approaches like CoTTA (26.63s) and SimATTA (45.45s).
The runtime profile demonstrates that BATTA-RL achieves a favorable balance between computational cost and performance, particularly considering its significant accuracy improvements over faster baselines while maintaining substantially lower processing time than methods like SimATTA.

| Method | Src | BN-Stats | TENT* | EATA* | SAR* | CoTTA* | RoTTA* | SoTTA* | SimATTA* | BATTA-RL |
|---------|-----|-----------|--------|--------|-------|---------|---------|---------|------------|------------|
| Avg. Time (s) | 0.18 | 0.33 | 1.03 | 0.98 | 1.02 | 26.63 | 1.68 | 1.25 | 45.45 | 4.19 |

Table R3. Average wall-clock time per batch (s) comparisons.

---

> ### Author Response · Authors · 2024-11-20
> **Global response (Part II)**
>
> ### Revision
>
> The revisions made are marked with “$\text{\color{blue}blue}$” in the revised paper.
>
> - Section 3.2: Design considerations on the reward function (Reviewer ZvDw).
> - Section 4: Additional results on datasets (Reviewer uYWH and afeQ), additional analysis on the runtime (Reviewer 8qrn, Pxu6, and afeQ)
> - Figure 7: Additional results on the number of active samples per batch (Reviewer afeQ).
> - Appendix B: Additional analysis on the number of epochs (Reviewer afeQ), use of augmentation (Reviewer Pxu6), and intermittent labeling (Reviewer afeQ).
> - Appendix C: Additional results on scenarios (Reviewer uYWH).
> - Appendix D.1: Explanations on TTA with binary feedback (Reviewer ZvDw).
>
>
> We are confident these revisions have significantly improved the paper's clarity and technical depth. We look forward to any additional feedback that would further strengthen our contribution.
>
> Sincerely,
>
> Authors

---

### Meta-Review · Area_Chair_FU9A · 2024-12-17

**Metareview:**

The paper addresses the challenge of adapting deep learning models to test-time domain shifts with minimal labeling costs. It introduces Binary-Feedback Active Test-Time Adaptation (BATTA), where an oracle (human) provides binary feedback (correct/incorrect) on selected predictions instead of full-class labels.

While this paper presents an interesting and novel angle with merits in its use of RLHF to update the model, the AC agrees with Reviewer 8qrn that ATTA could benefit significantly from leveraging a stronger foundation model. This includes not only using an advanced foundation model as an oracle to generate RL rewards but also at least experimenting with a larger teacher model to periodically guide the model at test time and measure the resulting performance gains. Although the authors state that exploring foundation models as oracles is a direction for future work, this investigation would greatly enhance the current paper by providing a deeper understanding of the benefits of human-in-the-loop approaches.

**Additional Comments On Reviewer Discussion:**

Although the paper presents an interesting idea and the rebuttal addresses some concerns from several reviewers, the AC still sees major issues, e.g., insufficient experimental evaluation. It is suggested that the authors address the remaining concerns to improve the quality of their submission.

---

### Decision · Program_Chairs · 2025-01-22

Reject